# Comprehensive proteolytic profiling of *Aedes aegypti* mosquito midgut extracts: Unraveling the blood meal protein digestion system

Anthony J. O'Donoghue[1], Chenxi Liu[1], Carter J. Simington[2], Saira Montermoso[3], Elizabeth Moreno-Galvez[3], Mateus Sá M. Serafim[1,4], Olive E. Burata[3], Rachael M. Lucero[3], James T. Nguyen[3], Daniel Fong[3], Khanh Tran[3], Neomi Millan[3], Jamie M. Gallimore[3], Kamille Parungao[3], Jonathan Fong[3], Brian M. Suzuki[1], Zhenze Jiang[1], Jun Isoe[2], Alberto A. Rascón, Jr[3]¤*

1 Center for Discovery and Innovation in Parasitic Disease, Skaggs School of Pharmacy and Pharmaceutical Sciences, University of California, San Diego, La Jolla, California, United States of America, 2 Department of Chemistry and Biochemistry, The University of Arizona, Tucson, Arizona, United States of America, 3 Department of Chemistry, San José State University, San José, California, United States of America, 4 Department of Microbiology, Federal University of Minas Gerais, Belo Horizonte, Minas Gerais, Brazil

¤ Current address: School of Molecular Sciences, Arizona State University, University Dr. Tempe, Arizona, USA
* aarasco1@asu.edu

**Data Availability Statement:** All relevant data are within the manuscript and its Supporting Information files. Additionally, the docking results

## Abstract

To sustain the gonotrophic cycle, the *Aedes aegypti* mosquito must acquire a blood meal from a human or other vertebrate host. However, in the process of blood feeding, the mosquito may facilitate the transmission of several bloodborne viral pathogens (*e.g.*, dengue, Zika, and chikungunya). The blood meal is essential as it contains proteins that are digested into polypeptides and amino acid nutrients that are eventually used for egg production. These proteins are digested by several midgut proteolytic enzymes. As such, the female mosquito's reliance on blood may serve as a potential target for vector and viral transmission control. However, this strategy may prove to be challenging since midgut proteolytic activity is a complex process dependent on several exo- and endo-proteases. Therefore, to understand the complexity of *Ae. aegypti* blood meal digestion, we used Multiplex Substrate Profiling by Mass Spectrometry (MSP-MS) to generate global proteolytic profiles of sugar- and blood-fed midgut tissue extracts, along with substrate profiles of recombinantly expressed midgut proteases. Our results reveal a shift from high exoproteolytic activity in sugar-fed mosquitoes to an expressive increase in endoproteolytic activity in blood-fed mosquitoes. This approach allowed for the identification of 146 cleaved peptide bonds (by the combined 6 h and 24 h blood-fed samples) in the MSP-MS substrate library, and of these 146, 99 (68%) were cleaved by the five recombinant proteases evaluated. These reveal the individual contribution of each recombinant midgut protease to the overall blood meal digestion process of the *Ae. aegypti* mosquito. Further, our molecular docking simulations support the substrate specificity of each recombinant protease. Therefore, the present study provides key information of midgut proteases and the blood meal digestion process in mosquitoes, which may be exploited for the development of potential inhibitor targets for vector and viral transmission control strategies.

are available in the Zenodo repository (under the code: 10.5281/zenodo.14605478).

**Funding:** Research reported in this publication was supported by the National Institute of General Medical Sciences (NIGMS) of the National Institutes of Health (NIH) under Award Number SC3GM116681 to AAR while at San Jose State University and R21AI180325 sub-award to AJO (PI, H. Jiang). The content is solely the responsibility of the authors and does not necessarily represent the official views of the National Institutes of Health. The funder had no role in study design, data collection and analysis, decision to publish, or preparation of the manuscript.

**Competing interests:** The authors have declared that no competing interests exist.

## Author summary

The *Aedes aegypti* mosquito is a vector of viral pathogens that can be transmitted directly to humans. For instance, the transmission of dengue, Zika, or chikungunya viruses may happen during the *Ae. aegypti* acquisition of an infected blood meal. This blood meal is important for the anautogenous mosquito because without the digestion of blood proteins the mosquito will not obtain the necessary nutrients needed for egg production. After imbibing a blood meal, midgut digestive enzymes (proteases) are expressed and secreted into the lumen. To fully understand their roles in blood meal digestion, we used a special technique called Multiplex Substrate Profiling by Mass Spectrometry (MSP-MS). This method allows us to generate global proteolytic activity profiles of *Ae. aegypti* midgut tissue extracts that were fed with sugar or blood. In addition, we generated substrate cleavage profiles of recombinantly expressed midgut proteases allowing us to understand the enzyme preferences for blood proteins. Therefore, utilizing this approach, we found the contribution of each individual recombinant protease tested relative to the global activity profile of blood-fed midgut tissue extracts. This may be a starting point for the validation of midgut protease inhibition and the development of a new vector control strategy.

## Introduction

The female *Aedes aegypti* mosquito heavily relies on a blood meal from a human host to help sustain the gonotrophic cycle. The blood meal contains proteins (hemoglobin, serum albumin, and immunoglobulin) that are digested by midgut proteases into polypeptides and amino acid nutrients used for energy production, egg development, and to replenish maternal reserves [1,2]. In fact, there is enough protein from one blood meal to produce more than 100 viable eggs [3]. Unfortunately, the blood feeding behavior of these mosquitoes has facilitated the transmission of several bloodborne viral pathogens, such as dengue, Zika, and chikungunya viruses (DENV, ZIKV, and CHIKV). Alarmingly, the Zika outbreak between 2015–2018 in the Western hemisphere has led both *Ae. aegypti* and *Ae. albopictus* in becoming well established primary and secondary vectors of the virus, respectively [4]. This is particularly troublesome for areas in the Southern United States, such as Florida, where the *Ae. aegypti* mosquito has reemerged leading to locally acquired transmission of DENV, ZIKV, and CHIKV [5–7]. The only strategy still in place to minimize and prevent viral pathogen transmission is through vector control, mostly utilizing insecticides [8]. However, resistance to commercially available pyrethroids and organophosphates has been reported for *Ae. aegypti*, which are the most common insecticides used to control mosquito populations in endemic areas [9]. Alternatively, the introduction of sterile insects into the population has its own logistical issues in effectively producing and releasing sterilized males [10]. Similarly, the release of *Wolbachia* infected mosquitoes, which have shown to suppress viral replication, has its own logistical constraints that limit the use of this biological method to prevent virus transmission [8,11]. Therefore, there is still a need for exploration of new and effective vector control strategies to minimize the mosquito population and viral pathogen transmission.

The female mosquito's reliance on a blood meal may prove to be an excellent target for mosquito population control [2,12]. This whole process takes about 48 hours, but for digestion to be initiated, the mosquito must consume more than her weight in blood, leading to the expansion of the mosquito midgut to fully accommodate the large uptake of blood. Upon the

imbibing of a blood meal, early phase midgut proteases are produced to initiate the digestion of blood proteins. This is followed by a second phase of midgut proteases after 12 h post-blood feeding, that are required to fully degrade blood proteins into the necessary nutrients required to fuel the gonotrophic cycle [1,13]. These enzymes may be targeted for the discovery of inhibitors, aiming to cease blood digestion and ultimately be employed as an alternative to control the mosquito population. However, such strategy may be challenging, as proteolytic activity is a complex process dependent on endo-proteases, amino-peptidases and carboxy-peptidases [1,14]. More specifically, the *Ae. aegypti* Early Trypsin (AaET) midgut protease was shown to be involved in the early phase of protein digestion ($\leq$ 6 h post blood feeding) [15,16]), while those involved in the later phase are *Ae. aegypti* Serine Protease VI (AaSPVI), VII (AaSPVII), and late trypsin (AaLT) [15–18]. These four proteases are the most abundantly expressed midgut proteolytic enzymes found in the mosquito (based on mRNA and protein expression profiles [1,15,18,19]), and were chosen for RNAi knockdown studies to identify which of these are directly involved in fecundity. Surprisingly, only the knockdown of AaSPVI, AaSPVII, and AaLT led to a significant reduction in the number of eggs laid by the *Ae. aegypti* mosquito compared to the FLUC (firefly luciferase) control (~22.9% overall reduction for each knockdown), while the knockdown of AaET had no effect on fecundity [1]. When all three late phase proteases were simultaneously knocked down only 31.2% reduction was observed. This lack of synergy indicated that other midgut proteases are likely to compensate for these missing enzymes. Later, another early phase protease named *Ae. aegypti* Serine Protease I (AaSPI) and four constitutively expressed midgut proteases (AaSPII-AaSPV) were identified, which are likely to be involved throughout the early and late phases of blood meal digestion [14]. These proteases may be involved in digesting blood meal proteins in the *Ae. aegypti* mosquito midgut but have not been isolated to confirm this. Finally, two other midgut proteases have been described, named AaCHYMO [20] and Juvenile Hormone Associated 15 (JHA15) [21], both of which are predicted to have chymotrypsin-like activity. In total, 11 different proteases are predicted to be present in the mosquito midgut during blood meal digestion, but it is still unclear what role each enzyme plays in this process.

Multiplex Substrate Profiling by Mass Spectrometry (MSP-MS) is a peptide digestion assay [22] and was previously validated as a method to uncover the global proteolytic activity in plasma [23]. Herein, MSP-MS was used to create a global proteolytic profile of *Ae. aegypti* midgut tissue extracts from sugar-fed and blood-fed mosquitoes to fully understand the proteolytic complexity of the blood meal digestion process (**Fig 1**). In addition, we used MSP-MS to generate in-depth substrate profiles of a select panel of recombinantly expressed midgut proteases to uncover the contribution of each enzyme to the global proteolytic activity of the midgut. The most abundant proteases (AaET, AaSPVI, AaSPVII, and AaLT), as well as (AaCHYMO) (which also has high mRNA and protein expression levels post-blood feeding [20]) were selected. Our results reveal that exo-protease activity is dominant in sugar-fed mosquitoes while there is a shift to endo-proteolytic activity in blood-fed mosquitoes. Further, we determined that peptides were most frequently cleaved on the C-terminal side of Arg and Lys residues and therefore the proteases responsible for this cleavage can be classified as having trypsin-like specificity. Additional cleavages on the C-terminal side of Tyr and Phe were also detected, which indicates that chymotrypsin-like enzymes are also active. When the midgut protease activity was compared to several recombinant midgut enzymes, we revealed which enzymes are likely to contribute to the digestion system of the mosquito. Lastly, the substrate specificity is supported by molecular docking simulations. Taken together, this paves the way for the full understanding of the function of all mosquito midgut proteases without the need to recombinantly express and characterize every enzyme in the digestive system. Finally, we provide key information of several potential inhibitor targets that could be exploited for future vector control strategies.

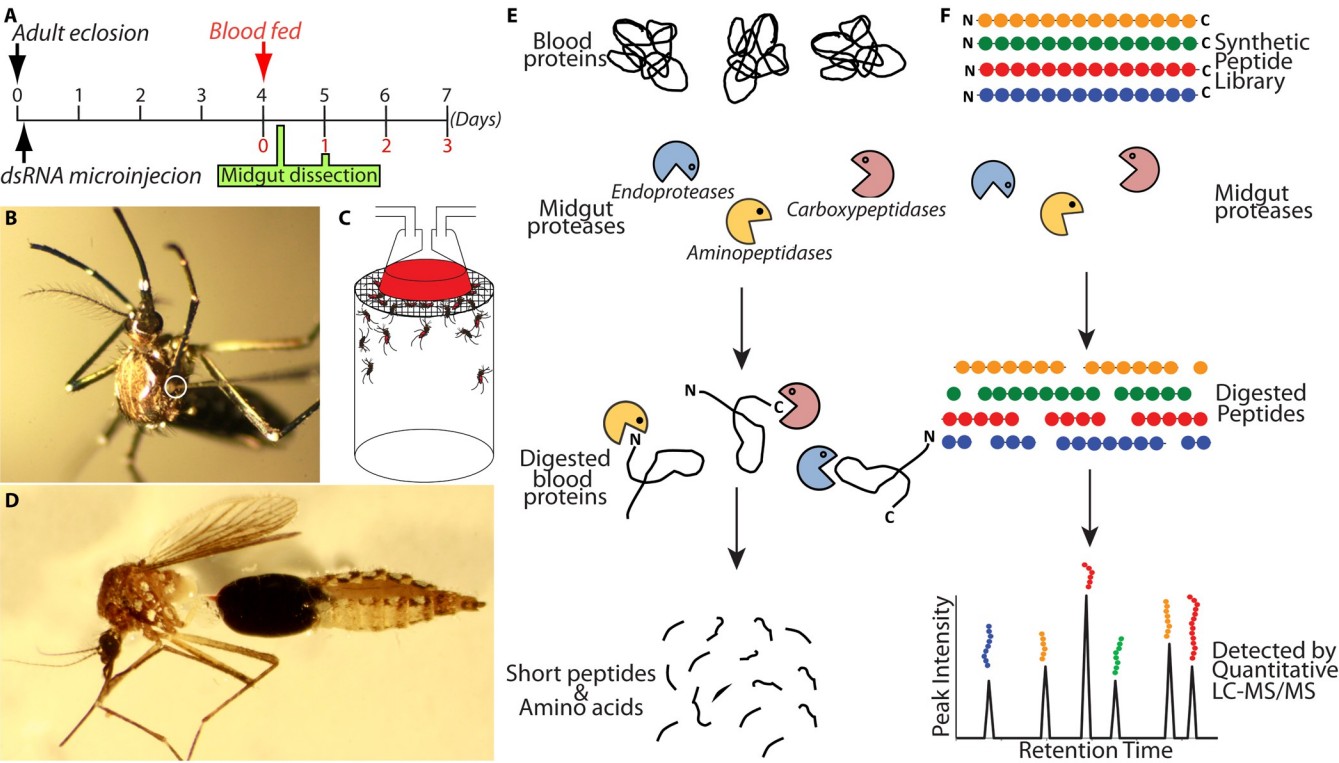

**Fig 1. Utilizing MSP-MS as an Approach to Create a Global Proteolytic Profile of *Aedes aegypti* Mosquito Midgut Tissue Extracts.** A) Schematic diagram of the experimental time course for dsRNA microinjection, blood feeding, and midgut dissection in the first gonotrophic cycle. B) dsRNA microinjection was performed within 4 h of adult eclosion using a Nanoject II micro-injector (Drummond Scientific Company, Broomall). The tip of a glass needle was inserted into the thorax for dsRNA injection (white circle). C) A schematic drawing showing a blood feed setting. Blood is kept warm (40˚C) in the glass feeder through the flow of circulating water. D) Midgut dissection of mosquitoes was performed in 1X PBS at 6- and 24-hours post-blood meal (PBM) under a light microscope. E) Schematic representation of blood protein digestion by midgut proteases yielding short peptides and amino acids. F) Schematic representation of the MSP-MS peptide library when incubated with midgut extracts containing proteases. Cleaved peptides are detected and quantified by liquid chromatography tandem mass spectrometry (LC-MS/MS). Images in A, C, E, and F are original and prepared by the authors. Adobe illustrator was used to draw these images. Dr. Jun Isoe took and prepared the images in B and D.

## Methods

### Mosquito rearing

For all experiments, the Rockefeller strain *of Ae. aegypti* mosquitoes were used. The mosquitoes were maintained on a 10% sucrose diet until blood feeding. Rearing conditions were set at 25˚C, 80% relative humidity, and a 16 h light: 8 h dark cycle.

### Blood feeding and midgut tissue preparation

Human blood was kindly donated by the American Red Cross and used to feed *Ae. aegypti* female mosquitoes. Parafilm was stretched over glass bells. Warded up to 37˚C. Mosquito midguts were then dissected in 1X PBS under a light microscope and transferred to 20 mM Tris-HCl pH 7.2 buffer. A total of 20 midguts in 200 µL buffer were prepared for each wildtype and RNAi sugar-fed and blood-fed samples used.

### Synthesis of Double-strand RNA (dsRNA)

Genes encoding *Ae. aegypti* serine proteases were identified from the genome in the NCBI database. The T7 promoter sequence (5' TAATACGACTCACTATAGGGAGA 3') was added

**Table 1. Gene-specific primers used for RNAi and qPCR.**

| Gene names | | Primer sequence (5' to 3') | PCR size |
|---|---|---|---|
| Gene-specific primers used for RNAi | | | |
| *Early trypsin, ET* | Forward | CCCTTCTGGGACTCAGCCA | 539 bp |
| | Reverse | TCCGTCACCGGAACAACGTT | |
| *Late trypsin, LT* | Forward | GCATCTCTGATGGCTTTGGCTTC | 516 bp |
| | Reverse | GCATTCGTTGTTACTGATCACCGT | |
| *5G1, AaSPVI* | Forward | CTCACAGCAACTTTCTTCGCC | 615 bp |
| | Reverse | CCTTCCTTGAATCCAGCGCAGA | |
| *CxLT, AaSPVII* | Forward | TGGCTCGTATCATCCTTCTGTT | 480 bp |
| | Reverse | CGAAACCTGCAGCAAAGTACC | |
| *Firefly Luciferase, Fluc* | Forward | AGCACTCTGATTGACAAATACGA | 548 bp |
| *pGL3-Basic Vector* | Reverse | AGTTCACCGGCGTCATCGTC | |
| Gene-specific primers used for qPCR | | | |
| *Early trypsin, ET* | Forward | ACCGTGGCAGATGGAGCTATG | 177 bp |
| | Reverse | GGCATAACCAGCGCAGATCAT | |
| *Late trypsin, LT* | Forward | GGAAGTGATACCTTTACCGACCG | 150 bp |
| | Reverse | GATCACCAACGGGCTGTAGGC | |
| *5G1, AaSPVI* | Forward | AGGAATGCCACAAGGCTTACTTGA | 156 bp |
| | Reverse | CCATAACCCCAGGATACCACT | |
| *CxLT, AaSPVII* | Forward | CCGCAGTACAACCCATCCAC | 146 bp |
| | Reverse | GATTCCGAAGGGTTTTGAGTATATC | |
| *Ribosomal protein S7, RPS7* | Forward | ACCGCCGTCTACGATGCCA | 131 bp |
| | Reverse | ATGGTGGTCTGCTGGTTCTT | |

T7 promoter sequence (5' TAATACGACTCACTATAGGGAGA 3') was added in 5' of each RNAi primer.

to the 5'-end of both the forward and reverse primers for each serine protease gene (**Table 1**). PCR was performed to amplify dsRNA region of DNA. The dsRNA's targeting each gene were synthesized using the NEB HiScribe T7 RNA Synthesis kit with PCR amplified DNA templates. The purified dsRNA was resuspended in nuclease-free HPLC water at 7.5 μg/μL and stored at -80˚C until use. *Ae. aegypti* females were microinjected with 2.0 μg dsRNA using a Nanoject II micro-injector (Drummond Scientific Company, Broomall). The injection was performed within 4 hours after adult eclosion. The dsRNA microinjected mosquitoes were maintained on 10% sucrose until needed for experiments. Immediately after dsRNA microinjection, mosquitoes were allowed to feed on 10% sucrose until dissection (see **Fig 1**). 90–95% RNAi knockdown efficiency of each protease target was achieved from 12 individual mosquitoes by qPCR using firefly luciferase control dsRNA microinjection.

## Determination of RNAi-Mediated Knockdown Efficiency of Protease Genes at the mRNA Level

The single mosquito qPCR method for knockdown verification has been previously described ([24,25]). Briefly, knockdown efficiency was verified by quantitative real-time PCR (qPCR) using protease gene-specific primers purchased from Integrated DNA Technologies (Table 1). Midgut tissues from RNAi-ET mosquitoes were dissected in 1X PBS at 3 hours PBM, while RNAi-LT, AaSPVI, and AaSPVII midguts were dissected at 24 hours PBM. Total RNAs were isolated using TRIzol (Invitrogen), and complementary DNA was synthesized from 200 ng of DNase I-treated total RNA using an oligo-(dT)-VN primer and reverse transcriptase

(Promega). qPCR was performed with Thermo Scientific Maxima SYBR Green qPCR Master Mix with a final primer concentration of 200 nM using the BioRad CFX Connect qPCR instrument. Relative expression for ET, LT, AaSPVI, and AaSPVII were normalized to ribosomal protein S7 (RPS7) transcript levels in the same cDNA samples. Knockdown efficiency was compared using FLUC dsRNA-injected mosquitoes as a control. Data was obtained from 12 individual mosquitoes. Data are presented as mean ± SEM of 12 individual mosquitoes. Statistical significance is represented by stars above each comparison (unpaired Student's *t* test; **** P < 0.0001 compared to the RNAi-FLUC control).

## Peptide cleavage site identification by msp-ms and proteolytic activity of mosquito midgut extracts and recombinant proteases

The tetradecapeptide library consists of 228 rationally designed peptides that are each 14 residues in length and have an equal distribution of 19 amino acids (no cysteine or methionine, but norleucine included), detailed information on the design of the peptide library is found in previous publications [22,26]. All neighbor and near-neighbor pairwise combinations of these 19 amino acids are present in the library such that the total number of cleavable bonds is 2,964 (13 bonds x 228 peptides). The peptides were mixed at an equal molar concentration and diluted in Dulbecco's phosphate-buffered saline (D-PBS) to a concentration of 1 μM for each peptide. Midgut samples (sugar-fed, 6 h and 24 h post-blood fed stocks) were diluted 50-fold in D-PBS, and 30 μL of each diluted sample was added to an equal volume of peptide mixture and incubated at room temperature. 10 μL aliquots were removed after 15- and 60-min incubation and protease activity quenched with 8 M Guanidinium hydrochloride. Please note that a blood only control MSP-MS assay was not performed, given that we previously published a study on blood (plasma) proteases revealing the activity in dominated by aminopeptidases and carboxypeptidases [23]. Recombinant enzymes were diluted in D-PBS such that the final concentration was 50 nM. A control sample consisted of midgut sample mixed with 8 M guanidinium hydrochloride prior to the addition of peptides. The assay was performed in four technical replicates. Samples were desalted with C18 tips and injected into a Q-Exactive Mass Spectrometer (Thermo Fisher Scientific, Waltham, MA, USA) equipped with an Ultimate 3000 HPLC. Peptides were separated by reverse phase chromatography using a C18 column (1.7 μm bead size, 75 μm × 25 cm, 65°C) at a flow rate of 300 nL/min with a linear gradient of solvent B (0.1% formic acid in acetonitrile) from 5% to 30% with solvent A (0.1% formic acid in water). Survey scans were recorded over a 150–2000 m/z range (70,000 resolutions at 200 m/z, AGC target $3 \times 10^6$, 100 ms maximum).

Tandem mass spectrometry (MS/MS) was performed in data-dependent acquisition mode with higher energy collisional dissociation (HCD) fragmentation (28 normalized collision energy) on the 12 most intense precursor ions (17,500 resolutions at 200 m/z, AGC target $1 \times 105$, 50 ms maximum, dynamic exclusion 20 s). The data were searched against tetradecapeptide library sequences, and a decoy search was conducted with sequences in reverse order with no protease digestion specified. Data were filtered to 1% peptide and protein level false discovery rates with the target–decoy strategy. Peak integration and data analysis were performed using Peaks software (Bioinformatics Solutions, Inc.). Peptides were quantified with label free quantification, and data were normalized by LOWESS and filtered by 0.3 peptide quality. Missing and zero values were imputed with random normally distributed numbers in the range of the average of the smallest 5% of the data ± standard deviation (SD). Missing and zero values are imputed with random normally distributed numbers in the range of the average of the smallest 5% of the data ± SD. Cleaved peptide products are defined as those with intensity scores of eightfold or more above the peptide intensity scores in the inactivated

**Table 2. AaCHYMO Forward and Reverse primers used for amplification of the gene of interest without the natural signal leader sequence [12].** The melting temperature ($T_M$) of each primer was estimated using NetPrimer (Premier Biosoft, Palo Alto, CA). The primers were purchased from ELIM Biopharmaceuticals, Inc. (Hayward, CA). The restriction enzymes for each primer are underlined and in bold.

| Gene Construct | Primer | Primer Sequence | $T_M$ (˚C) |
|---|---|---|---|
| AaCHYMO Zymogen (WT) No Leader | AaCHYMO-Zym-pET-Fwd No Leader | 5'-AAAAA**CATATG**ACCCACAAGATCGTCGGTGG-3' | 67.68 |
| | AaCHYMO -Zym-pET-Rev 1 | 5'-AAAAAA**AAGCTT**ATTACGCACGGAGCTGCTGCT-3' | 64.26 |

enzyme sample and q value < 0.05. Raw mass spectrometry data can be found on the MassIVE server (https://massive.ucsd.edu/ProteoSAFe/static/massive.jsp) and searched with the following identifiers: MSV000094822 (Sugar fed & blood fed MSP-MS data) and MSV000094823 (*Ae. aegypti* serine proteases). To validate the enzyme activity, 5 µM of Benzoyl-Phe-Val-Arg-7-amino-4-methylcoumarin (Bz-FVR-AMC) was assayed with a 1 in 1000 dilution of midgut extract (sugar-fed, 6 h and 24 h post-blood fed) or human blood in 20 mM Tris-HCl, pH 7.5, 100 mM NaCl, 10 mM $CaCl_2$, 0.01% Tween-20 at room temperature. Fluorescent readings at excitation 360 nm and emission 460 nm were taken on a BioTek Synergy HTX in 10.6 min intervals. Data for three technical replicates were recorded.

## Engineering of recombinant protease constructs

The AaET, AaSPVI, AaSPVII, and AaLT genes were cloned into the pET vector system, as described in [12]. AaCHYMO (NCBI Accession # U56423.1) was cloned into the pET28a vector (Novagen Cat. #69864–3) using the NdeI and HindIII restriction sites (**Table 2**). SignalP 4.1 [27] was used to identify the signal leader sequence of AaCHYMO for removal during PCR amplification. The AaCHYMO primers (**Table 2**) and purified *Ae. aegypti* mosquito cDNA (prepared as described in [28]) were incubated in GoTaq Green Master Mix (Promega, Cat. #M7122, Madison, WI), amplified using an Eppendorf Mastercycler, following the GoTaq Green Master Mix manufacturer's protocol with an annealing temperature of 64˚C (20 sec). Once the AaCHYMO plasmid was engineered, the construct was sent to ELIM Biopharmaceuticals, Inc. (Hayward, CA) for DNA sequencing and sequenced verified using NCBI BLAST.

## Soluble Expression, Purification, and Activation of Recombinant Midgut Proteases

All protease pDNA constructs (25 to 50 ng pDNA) were transformed into SHuffle T7 Express Competent *E. coli* Cells (New England Biolabs #C3029J) and plated on LB Agar kanamycin (30 µg/mL) plates, following the manufacturer's protocol. For large scale soluble expression of each protease, the protocol described in [12] was followed. Briefly, a single colony from a transformed plate was selected to set overnight cultures in LB media supplemented with kanamycin (30 µg/mL) grown for 16–18 h at 30˚C (250 rpm). The next morning large scale growths (500 mL in 1 L Erlenmeyer flasks, total between 3 to 6 L) in terrific broth (TB) media (for AaET, AaSPVI, AaSPVII, and AaLT) and LB media (for AaCHYMO) supplemented with kanamycin (30 µg/mL) were set using the overnight starter culture (starting at an $OD_{600nm}$ ~ 0.05) at 30˚C (250 rpm). The growth for each was monitored, and at an $OD_{600nm}$ ~ 0.5–1.0, cells were induced with 0.1 mM IPTG, followed by the reduction of the growth temperature to slow down bacterial growth, and promote recombinant protease folding [12]. AaET was grown at 23˚C for 70 min, AaSPVI at 15˚C for 18 h, AaSPVII at 23˚C for 6 h, AaLT at 10˚C for 48 h, and AaCHYMO at 12˚C for 30 h. Cell paste was collected and flash frozen in liquid nitrogen until needed for purification.

## Nickel purification of zymogen proteases

Given that all recombinant proteases have either a C- or N-terminal his$_6$-tag, the proteases were Nickel purified using 20 mM Tris-HCl pH 7.2 + 250 mM NaCl + 10 mM Imidazole + 2 mM DTT as the binding buffer and 20 mM Tris-HCl pH 7.2 + 250 mM NaCl + 500 mM Imidazole + 2 mM DTT as the eluting buffer using a triple gradient, as described in [12]. Once Nickel purified, AaET, AaSPVI, AaSPVII, and AaLT fractions were pooled together in Dialysis Membrane Tubing (10K NMWL Fisher Scientific, Cat. #08-667D) and dialyzed in 50 mM sodium acetate pH 5.2 + 150 mM NaCl + 1 mM DTT in a 2 L graduated cylinder at 4˚C, with fresh buffer exchanged the next day and dialyzed for another four hours to ensure removal of imidazole. AaCHYMO was similarly dialyzed, but in 20 mM Tris-HCl pH 7.2 + 10 mM CaCl$_2$ buffer, twice, as indicated. After dialysis, all proteases were cold centrifuged (4˚C) at 3500 rpm for 30 min to remove any precipitate, followed by concentrating using an Amicon Ultra-15 Centrifugal Filter Unit (Regenerated Cellulose 10,000 NMWL Ultracel-10K Millipore, Cat. #UFC901024) for 15 min intervals at 3,500 rpm at 4˚C until a final volume between 100 and 500 μL was achieved.

## Activation of Zymogens and further purification of activated proteases

For activation of the Nickel purified zymogen form of all proteases, the protease samples were set in dialysis buffer and buffer exchanged, followed by further purification using the FPLC. For AaET, the concentrated Ni$^{2+}$ purified protein was placed in dialysis membrane tubing and dialyzed in 2 L of 20 mM Tris-HCl pH 7.2 + 10 mM CaCl$_2$ for approximately 4.5 h at 4˚C. After activation, the protease was diluted to 10 mL with TRIS pH 7.2 and NaCl (final conditions of 20 mM Tris-HCl pH 7.2 + 500 mM NaCl (Buffer A)) in preparation for purification using a 5 mL HisTRAP Benzamidine column (GE Healthcare, Cat. #17-5144-01). The column was equilibrated using the AKTA Pure L1 FPLC (GE Healthcare), followed by loading of the activated protease sample, and eluted with a double gradient of 50% B for 10 CV and 100% B for 25 CV using a low pH elution buffer (10 mM HCl pH 2.0 + 500 mM NaCl (Buffer B)). Fractions were then collected and concentrated using the Amicon Ultra-15 centrifugal filter (10k NMWL). The protease was buffer exchanged and washed with 50 mM sodium acetate pH 5.2 + 100 mM NaCl (final buffer conditions). Once concentrated, active AaET was flash frozen in liquid nitrogen and stored until needed.

As for AaSPVI and AaSPVII, the proteases were activated by dialysis in 2 L of 20 mM Tris-HCl pH 7.2 + 10 mM CaCl$_2$ for approximately 16 to 18 h at 4˚C, with fresh dialysis buffer replaced halfway through. Activation was analyzed through SDS-PAGE and the synthetic trypsin substrate Nα-benzoyl-DL-arginine-4-nitroanilide (BApNA) (Acros Organics, Cat. #AC227740010) activity assays [12,28]. Both AaSPVI and AaSPVII were further purified using a MonoS column. After activation, the proteases were diluted to a final volume of 10 mL with ice-cold 1 M acetic acid pH 4.3 (final conditions were in 50 mM acetic acid pH 4.3). The proteases were then loaded onto an equilibrated MonoS 5/50 GL ion exchange column (GE Healthcare Cat. #17-5168-01) and washed with 10 column volumes (CV) of Buffer A (50 mM acetic acid pH 4.3), followed by a three-step linear elution gradient (10%B for 3 CV, 30%B for 6 CV, 50%B for 5 CV) with 50 mM acetic acid, pH 4.3 + 1 M NaCl (Buffer B). Fractions (150 μL each) containing the protease of interest were collected (detected by protein gel analysis) and pooled together. The pooled purified fractions were then concentrated using the Amicon Ultra-15 Centrifugal Filter (10 kDa NMWL), as described earlier. The final concentrated proteases were then aliquoted, flash frozen in liquid nitrogen, and stored at -80˚C.

AaCHYMO zymogen was also activated via dialysis, but with 20 mM Tris-HCl + 170 mM NaCl + 10 mM CaCl$_2$ and set at room temperature. In this case, however, Nickel purified

AaCHYMO was diluted to a total volume of 10 mL with 20 mM Tris-HCl pH 7.2 to a final concentration of 0.98 mg/mL. Activation was analyzed via SDS-PAGE gel analysis and final activity tested using the synthetic chymotrypsin substrate N-Succinyl-L-alanyl-L-alanyl-L-prolyl-L-phenylalanine 4-nitroanilide (Suc-AAPF-pNA) (Bachem Cat. #L1400.0050) dissolved in DMSO. Unlike all the other proteases, no further purification was required for AaCHYMO.

Unlike AaET, AaSPVI, AaSPVII, and AaCHYMO, AaLT does not auto-activate under any dialysis conditions tested and was therefore activated by incubating the $Ni^{2+}$ purified AaLT zymogen with activated and purified AaSPVI. To achieve this, AaLT was co-incubated with AaSPVI at a 13 μg AaLT to 1 μg AaSPVI ratio in 20 mM TRIS-HCl pH 7.2 + 10 mM $CaCl_2$ conditions (4.05 mL total volume) and incubated at room temperature for 10 min. After the 10 min incubation, the reaction mixture was diluted to a final volume of 10 mL with ice-cold 1 M acetic acid pH 4.3 in preparation for MonoS column purification (final conditions were 50 mM acetic acid pH 4.3). To isolate active AaLT, only 5 mL of the diluted reaction was purified at a time and loaded onto an equilibrated MonoS 5/50 GL ion exchange column and purified as described for AaSPVI and AaSPVII above. Fractions containing the protease of interest were collected and concentrated using an Amicon Ultra-15 Centrifugal Filter (10 kDa NMWL). The final concentrated protease was aliquoted, flash frozen in liquid nitrogen, and stored at -80˚C. The concentration of all proteases was estimated using the Pierce BCA Protein Assay Kit (Thermo Fisher Cat. #23227). To ensure AaLT was active, initial activity assays with Suc-AAPF-pNA was used, as described above for AaCHYMO. Lastly, all purified samples were loaded (~ 5 μg) onto a 4–12% BIS-TRIS gel (Invitrogen, Carlsbad, CA) in the presence of pre-stained PageRuler protein ladder (Thermo Scientific Cat. #26616) for final purity assessment and high-resolution separation.

### Molecular docking of preferred substrates with modeled midgut proteases

The midgut protease sequences (FASTA) were submitted to AlphaFold2 [29] via ColabFold (v1.5.5; https://github.com/sokrypton/ColabFold) [30] for modeling of their three-dimensional (3D) structures. Models were predicted with options set as default. Molecular docking with modeled midgut proteases was performed with GOLD 5.1 [31]. The fire ant trypsin (PDB ID: 1EQ9 [32]) co-crystallized with phenylmethylsulfonyl fluoride (PMSF), a common serine protease inhibitor, was used for validation and comparison purposes. The docking scoring function, the flexibility of residues and atoms, and the number of docking poses, were previously tested and evaluated to better reproduce the experimental binding mode of the co-crystallized ligand [33]. The redocking of the co-crystallized inhibitor considered root-mean-square deviation (RMSD) values $\leq$ 2.0 Å [34]. This provided the reproduced binding mode reference for validation of the docking protocol using GOLD. Protein structures and ligands were prepared as described in [35]. Ligands consisted of peptide sequences with four amino acids each, corresponding to their suggested P2-P1*P1′-P2′ positions: (i) QR*VI; (ii) YR*MI; (iii) FF*RL; (iv) IF*FR; and (v) MW*RL. Simulations were performed by building a grid box around the co-crystallized inhibitor of the reference structure, which was inserted into the modeled protease structures [36]. The parameters used in GOLD were: (i) all protein rotatable bonds were fixed; (ii) the binding site was defined from the inserted position, with all atoms within 10 Å [37]; (iii) gold_serine_protease VS was used as a template; (iv) 20 genetic algorithm runs were performed; (v) GoldScore was used as the scoring function [38], with no "early termination"; and (vi) the genetic algorithm search option was set as slow and "automatic". The representative pose was selected based on docking scores and by visual inspection [39], considering poses having the scissile bond prone to the nucleophile interaction to occur ($\leq$ 2.5 Å) ([40]). The PyMOL software (v2.5.7) was used to analyze the results and generate images.

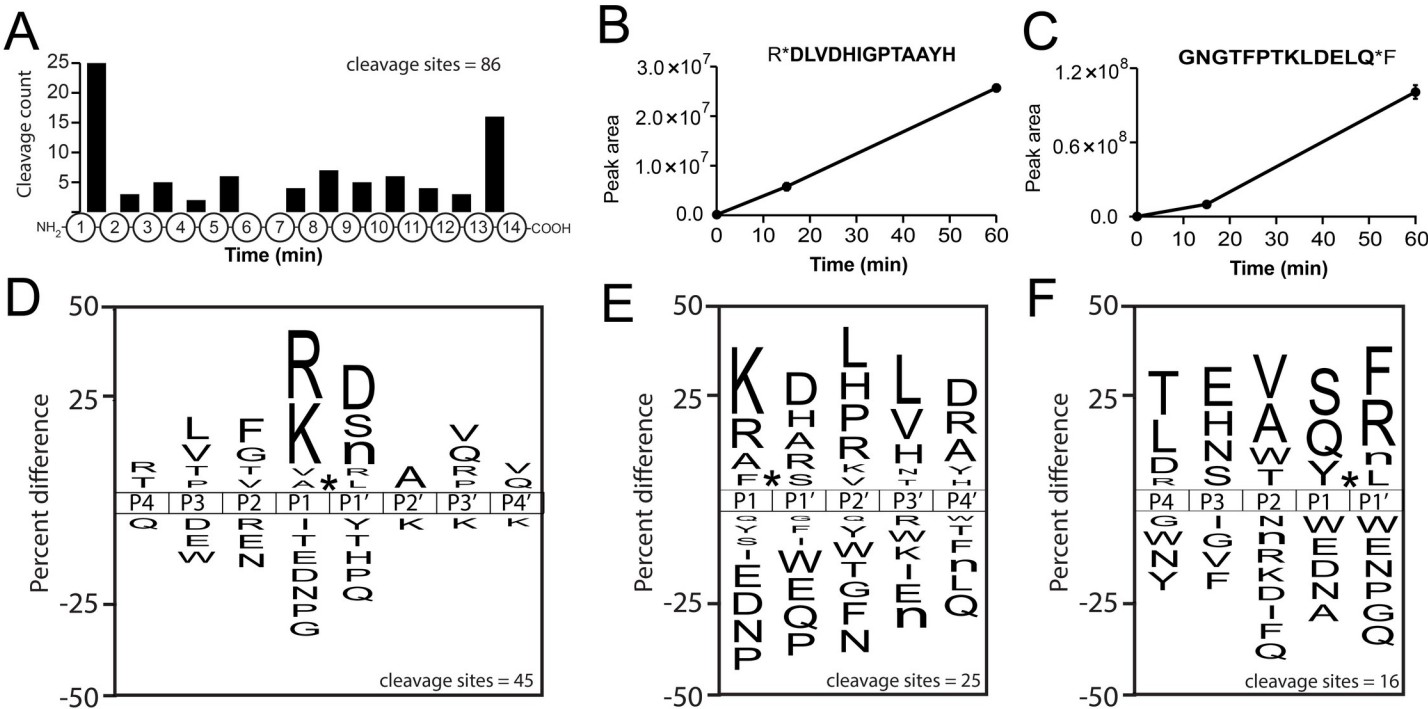

**Fig 2. Protease substrate profile of mosquito midgut following 6 hours of sugar feeding.** A) Distribution of cleavage sites within the 14-mer peptides. B) Example of a single peptide being cleaved by an aminopeptidase. The cleavage product in bold text was quantified by mass spectrometry and error bars correspond to the standard deviation (n = 3). C) Example of a single peptide being cleaved by a carboxypeptidase. Error bars correspond to the standard deviation (n = 3). The cleavage product in bold text was quantified by MS. Substrate profiles were represented using iceLogo for the cleavage sites of D) endopeptidases, E) aminopeptidases, and F) carboxypeptidases.

## Results

MSP-MS was used to uncover the proteolytic profile of control (sugar-fed) mosquito midgut extracts at 15- and 60-min post-incubation. After 60 min, we detected 86 cleaved peptides. Of note, 25 of these cleavage sites occurred at the N-terminus, while 16 occurred at the C-terminus (**Fig 2A**). This profile indicated the presence of aminopeptidase and carboxypeptidase activity. The remaining 45 cleavage sites were distributed between the 2nd and 13th amino acid of the 13-mer peptides. An example of a peptide cleaved by an aminopeptidase is illustrated in **Fig 2B** where RDLVDHIGPTAAYH is cleaved between Arg and Asp and the increase in the C-terminal product (DLVDHIGPTAAYH) can be quantified with time. For the carboxypeptidase activity, an example is shown using the substrate GNGTFPTKLDELQF that is cleaved after Gln to release the C-terminal Phe (**Fig 2C**). The remaining cleaved products can be found in **S1 File**.

We next generated a substrate specificity profile of the endo-proteases, amino-peptidases and carboxy-peptidase activity using iceLogo by comparing the frequency of amino acids surrounding the cleavage site. Using the 45 cleaved peptides that occurred between position 2 and 13, the protease responsible has a strong preference for Arg or Lys at the P1 position and a lower preference for Ala and Val. In addition, Asp, Ser and norleucine (Nle) were preferred at the P1′ position, Leu and Val at P3, Phe and Gly at P2, Ala at P2′, and Val and Gln at P3′ and P4′ (**Fig 2D**). Overall, the major substrate profile is considered "trypsin-like" since cleavage occurs after positively charged residues, however, a low level of elastase-like activity is also present due to cleavage after Ala and Val. Next, we generated a substrate profile of the

aminopeptidase activity. The enzymes responsible for these cleavages prefer Lys and Arg in the P1 position and Asp at P1′ and P4′, and Leu at P2′ and P3′ (Fig 2E). Finally, the carboxy-peptidases in the midgut extracts preferentially removed Phe and Arg from the carboxy terminus and Ser, Val, Glu and Thr were most frequently found at the P1, P2, P3 and P4 positions, respectively.

Using sugar-fed mosquito midguts as a control, we then compared the activity in midguts extracted from mosquitoes at 6 h and 24 h post human blood-fed with the same peptide library as described in Fig 2. The proteases in the 6 h blood-fed extract cleaved at 49 sites of which only 12 overlapped with the sites cleaved in the 6-h sugar-fed extracts, indicating a major change in midgut active proteases between these two conditions (Fig 3A). In 24 h blood-fed extracts the total number of cleaved peptide bonds increased to 180 which consisted of 109 uniquely cleaved sites. Only nine sites were commonly cleaved by proteases in the three different midgut extracts. Overall, these data reveal that functional proteolytic changes occur in the mosquito midgut at 6 h and at 24 h, which supports previous expression studies [1,14].

When looking closely at the distribution of the cleavages within the 14-mer peptide (Fig 3B), proteases in the 6 h and 24 h blood-fed extracts cleaved at only two and four sites, respectively, at the N-terminus (between amino acid 1 and 2) which compares to 25 peptides cleaved in the sugar-fed extracts. One peptide F*SLSKMNPVSQVLH was cleaved by proteases in all three samples and the peak intensity of the cleavage product was higher in the 6-hour sugar-fed extracts (Fig 3C). Two peptides (GNGTFPTKLDELQ*F and RDLVDHIGPTAAY*H) were cleaved at the C-terminus by proteases in both sugar-fed and blood-fed extracts. For the remaining endopeptidase cleavage sites, only six sites were commonly cleaved by all midgut samples. One such substrate, FSLSKnNPVSQVLH, was cleaved between Lys and Nle more efficiently by proteases in the blood-fed extracts relative to the sugar-fed extracts (Fig 3D). As FSLSKnNPVSQVLH is cleaved more efficiently by proteases in sugar-fed midguts between Phe and Ser, but less efficiently between Lys and Nle, indicates that there is a change in proteolytic activity between sugar-fed and blood-fed midguts.

A substrate specificity profile was generated of the endopeptidase activity in the 6 h and 24 h samples and revealed that both samples had a strong preference for Lys and Arg at P1. In addition, proteases in both samples were able to cleave peptides with P1-Phe with lower frequency than Arg or Lys. In the 6 h blood-fed profile, Val, Phe, and Lys are preferred in P2, Nle and His in P3, and Val, Glu and Ala in P1′ (Fig 3E). In the 24 h blood-fed profile, Arg is frequently found in P2′ while Phe, Nle and His are preferred in P1′ (Fig 3F).

To validate the activities seen in the midgut samples, we identified a fluorogenic substrate with the P3, P2 and P1 sequence of Phe-Val-Arg that also contains an N-terminal benzoyl capping group and a C-terminal fluorescent reporter group consisting of 7-amino-4-methylcoumarin (AMC). This substrate was not cleaved by proteases in the sugar-fed sample but was cleaved by proteases in blood-fed extracts (Fig 3G), with higher activity noted for the 24 h sample. As a control, Bz-FVR-AMC was incubated with human blood and no cleavage was detected, which confirmed that the protease activity in blood-fed midguts did not originate from blood.

We predicted that the protease(s) responsible for cleaving Bz-FVR-AMC was one of the serine proteases that are highly expressed in mosquito midguts following a blood meal. To confirm this, 6 h and 24 h midgut samples were pre-incubated with AEBSF, a broad-spectrum serine protease inhibitor and then assayed with Bz-FVR-AMC. Activity was decreased to almost undetectable levels by AEBSF which confirmed that the family of proteases responsible for the activity is "trypsin-like" (Fig 4A). As a control, bovine trypsin was also inactivated following pre-incubation with AEBSF. To uncover the serine protease responsible for the activity against Bz-FVR-AMC, AaET, AaSPVI, AaSPVII, and AaLT were knocked down using RNAi

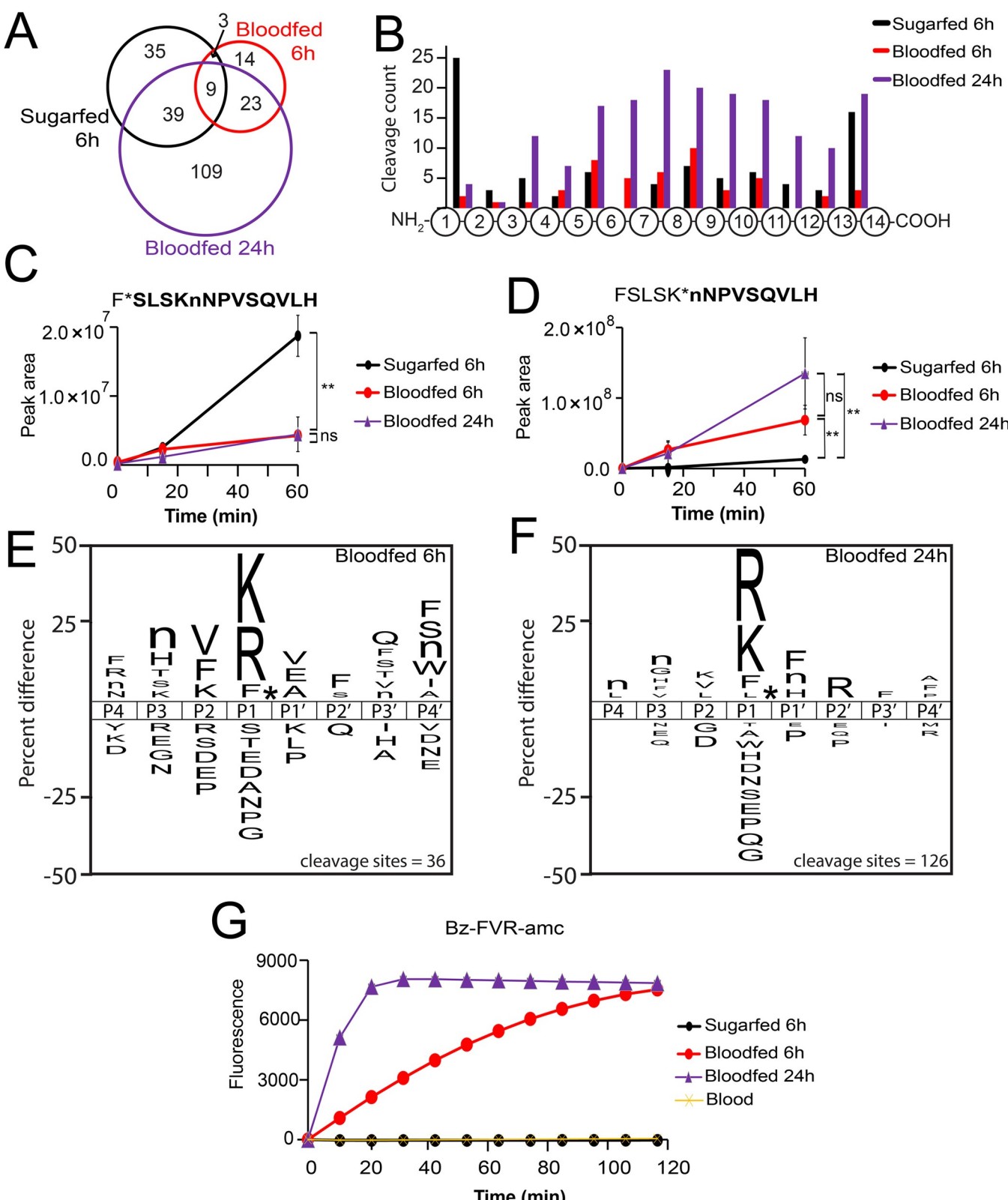

**Fig 3. Protease substrate profile of mosquito midgut following 6 and 24 hours of blood feeding.** A) Overlap of peptide cleaved sites by proteases in sugar-fed (black), 6 h blood-fed (red), and 24 h blood-fed midguts (purple). Only nine sites were commonly cleaved by proteases in the three protein extracts. B) Distribution of cleavage sites within the 14-mer peptides. C) F*SLSKMNPVSQVLH is an example of a substrate that was cleaved by proteases in all three

samples but is significantly higher (** P<0.05) in sugar-fed midguts after 60 mins. D) FSLSK*nNPVSQVLH is an example of a substrate that was preferentially cleaved by proteases in blood-fed midguts at a significantly higher rate (** P<0.05) than proteases in sugar-fed midguts. There is no significant (ns) difference between 6 h and 24 h protease activity. E) Substrate specificity profile of 6 h blood-fed midgut where lowercase n corresponds to norleucine. F) Substrate specificity profile of 24 h blood-fed midgut. G) Cleavage of Bz-FVR-AMC by midgut samples and human blood. Error bars are present at each timepoint (n = 3) but are too small to be visible.

and compared to that of the wildtype extract and a FLUC control. Here, only the knockdown of AaSPVI resulted in a significant decrease in trypsin-like activity when compared to FLUC (**Fig 4B**). RNAi-mediated knockdown efficiency was validated via qPCR, resulting in overall significant decrease in the mRNA levels of AaET, AaSPVI, AaSPVII, and AaLT (**Fig 4C**).

We next sought to uncover the substrate specificity of a selection of midgut proteases that are predicted to play key roles in blood feeding (AaET, AaSPVI, AaSPVII, AaLT and AaCHYMO) [1,14,20]. Recombinant proteins were purified on a nickel column, activated, and the mature enzyme was re-purified. The approach, as described in [12], resulted in near homogenous purified and active proteases (**Fig 5**).

MSP-MS was utilized to determine the proteolytic profile of each recombinant protease. 50 nM of each protease was incubated with 0.5 μM of each peptide of the 228-peptide library (114 μM total peptide) resulting in Arg and Lys preference at the P1 position for AaET, AaSPVI, and AaSPVII (**Fig 6A–6C**). Obvious differences at other P positions can be observed between the trypsin proteases, which leads to differences in protease cleavage specificity and preference. When looking closely at the similarities and differences in the cleavage of the 14-mer peptide substrates, AaET, AaSPVI, and AaSPVII cleave 34 peptides at the same site, with 27 unique substrates being cleaved by AaET, 16 unique cleaved by AaSPVI, and 14 cleaved by AaSPVII (**Fig 6D**).

The specificity profile of AaLT (**Fig 7A**) has some overlapping features with AaCHYMO (**Fig 7B**) in that it prefers Phe, Leu, and Tyr in the P1 position, however AaLT also prefers bulky amino acids such as Trp and His. Importantly, this protease did not cleave peptides with Arg or Lys at the P1 position. Beyond the P1 position, AaLT preferred to cleave peptides with hydrophobic amino acids at P4, P2′ and P4′, polar and positively charged amino acids at P3 and P1′. As for AaCHYMO, the specificity profile closely resembles the activity of chymotrypsin-like proteases (*e.g.*, bovine) with specificities towards Tyr, Leu, and Phe (**Fig 7A**) [41,42]. Ala and Asn are preferred at P4 and P4′, respectively, while Arg is most frequently found at the P2′ position. When comparing the peptides cleaved by AaCHYMO and AaLT, only 12 substrates were in common (**Fig 7C**).

With knowledge of the peptide bonds in the library that are cleaved by midgut proteases and by the recombinant enzymes, we were able to determine how much of the midgut protease activity could be assigned to the five enzymes. Firstly, we determined that 146 peptide bonds were cleaved by proteases in the midgut upon blood feeding (combination of 6 h and 24 h data). From these we revealed that AaSPVI cleaves 45 of these same bonds while AaSPVII cleaves 37 (**Fig 7D**). Overall, we revealed that the five recombinant enzymes were capable of cleaving 68% of the peptide bonds (99 out of 146). Therefore, the remaining 32% of cleaved peptides are likely due to other proteases in the midgut that we have not yet characterized.

## Substrate selectivity is supported by molecular docking simulations

Aiming to understand the observed amino acid preferences, we performed molecular docking simulations with selected substrates to the different midgut enzymes. Initially, the redocking of the co-crystallized inhibitor PMSF to the fire ant trypsin (PDB ID 1EQ9) using GOLD 5.1 (RMSD = 1.82 Å) validated the docking protocol (≤ 2.0 Å) [34]. Next, a grid box of 10 Å was

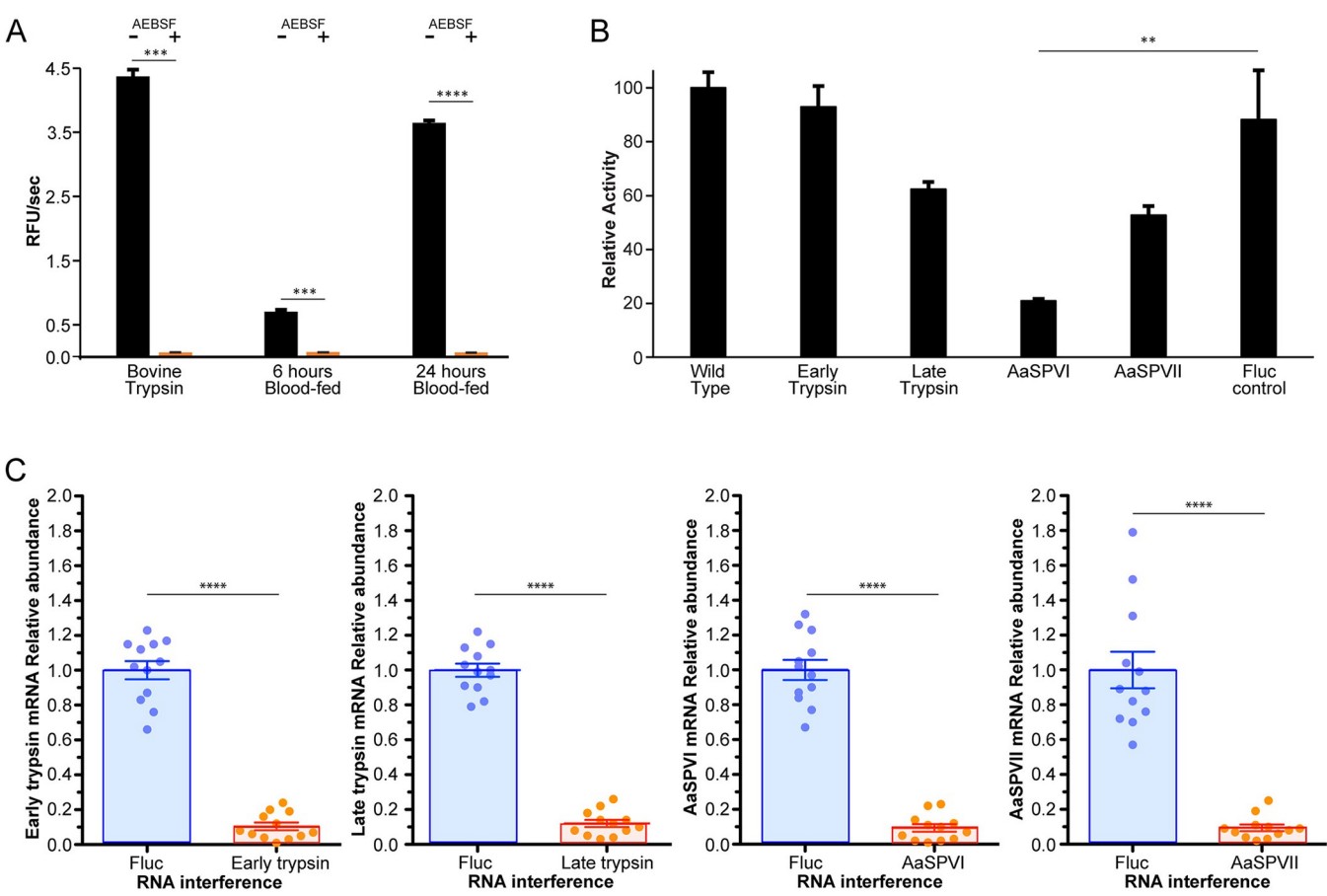

**Fig 4. Changes in midgut protease activity in response to chemical inhibition and gene expression knockdown.** A) Comparison of protease activity in the presence and absence of the serine protease inhibitor, AEBSF. B) Activity in wildtype midgut extracts compared with activity in midguts with proteases knocked down by RNAi. C) Relative mRNA levels of AaET, AaSPVI, AaSPVII, and AaLT. RNAi-mediated knockdown efficiency was measured by quantitative real-time PCR (qPCR). Individual mosquitoes were used to isolate total RNA, for cDNA synthesis, and to monitor silencing efficiency. Data are presented as mean ± SEM of 12 individual mosquitoes. **P<0.05, *** P<0.001, **** P < 0.0001 compared to the RNAi-FLUC control.

defined around the inhibitor inserted into the five modeled midgut protease structures. Our docking using GOLD predicted the expected QR*VI and YR*MI carbonyl groups within feasible distances for nucleophilic attack ($\leq 2.5$ Å) [40] of the catalytic serine of AaSPVI, AaSPVII, and AaET (**Fig 8**). Similar poses were predicted for the substrates among the three proteases, having potential P1 Arg contacts predicted with $Asp_{181}$ and $Gly_{180}$ backbones close to the catalytic serine in AaET (**Fig 8A and 8D**) and $Gly_{180}$ in AaSPVI (**Fig 8E**), while only $Phe_{26}$ and $His_{42}$ in AaSPVI with QR*VI (**Fig 8B**). As for AaSPVII predictions, the P1 Arg is less likely or does not interact with an aspartate or glycine, rather having potential contacts with $Phe_{25}$ and an additional $Trp_{197}$ interaction at P2′ (**Fig 8C and 8F**). On the other hand, potential contacts were predicted for both substrates with the catalytic histidine in AaSPVI at P1 and P2, respectively (**Fig 8B and 8D**), but with QR*VI having no predicted aspartate or glycine interactions to this target (**Fig 8B**). One could argue that these aspartate or glycine interactions could be important for the accommodation of the P1 Arg close to the catalytic serine. Also, P2 positions do not seem to be essential for interactions within the binding site. These results are consistent with the profiling of these substrates cleavage, showing a strong preference only for Arg at P1 (**Fig 6A–6C**) and highlighting a closer relation between AaET and AaSPVI rather than AaET and AaSPVII (**Fig 6D**). The predicted P1 contacts are summarized in **Fig 8**.

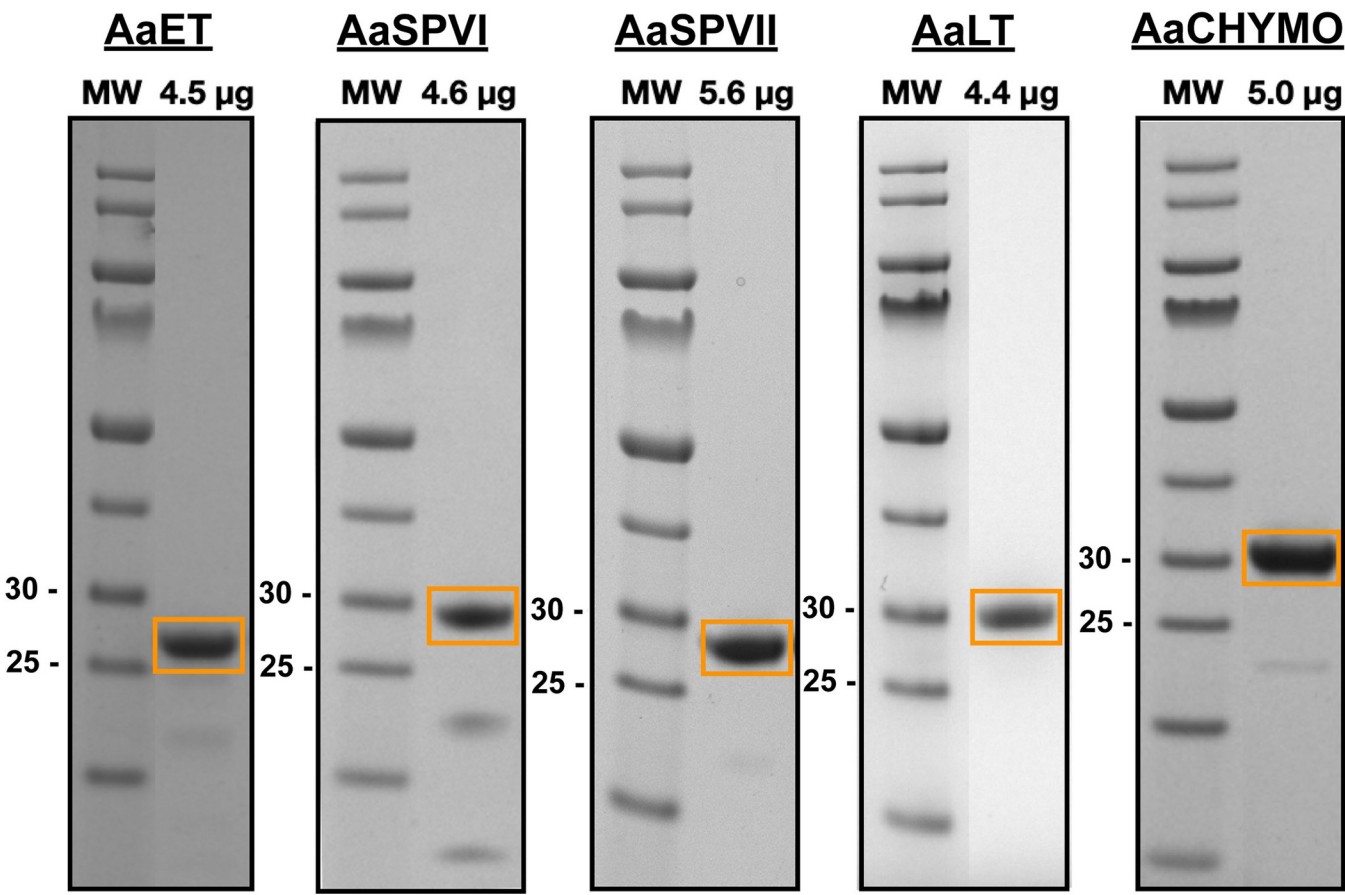

**Fig 5. Activated and purified recombinant midgut proteases used for MSP-MS analysis.** Purified proteases (4.4 to 5.6 µg) were loaded onto NuPAGE 4–12% BIS-TRIS gels (Invitrogen) to assess molecular weight and purity. All mature, active proteases are boxed in orange. MW represents the pre-stained PageRuler protein ladder in kilodaltons (kDa) (Thermo Scientific #26616 or #26619).

Further, we asked whether the specific substrate cleavage differences observed for AaCHYMO and AaLT (**Fig 7**) would also be supported by docking simulations. AaLT was able to cleave F*R and W*R, which are not cleaved by AaCHYMO. This protease cleaves F*F instead and has a stronger preference for Arg at P2′ (**Fig 7B**), while AaLT prefers Arg at P1′ (**Fig 7A**). Additionally, both enzymes do not cleave R*V and R*M (Arg at P1) and can accommodate Phe at P1, but AaCHYMO does not tolerate Trp at P1 (**Fig 7B**). These results are consistent with our docking simulations using GOLD, in which FF*RL (**Fig 9A**) and MW*RL (**Fig 9B**) are predicted to be within distances not feasible for the expected nucleophile attack ($\leq 2.5$ Å) in AaCHYMO or having poses with multiple steric clashes. As for IF*FR (**Fig 9C**), a potential π-stacking is predicted for the P1 Phe with the catalytic $His_{48}$, while the P2′ Arg is facing the opposite side of the catalytic $Ser_{193}$, interacting with $Ser_{190}$. Conversely, FF*RL (**Fig 9D**) was predicted to interact with AaLT similarly to MW*RL (**Fig 9E**). However, the latter was predicted to have a higher distance for the expected scissile bond to occur (~3.4 Å), which would be inconsistent with cleavage feasibility ($\leq 2.5$ Å). Still, a potential interaction of P1′ Arg with $Ser_{178}$ (**Fig 9E**) was predicted from our simulations, which is consistent with the biochemical data (**Fig 7A**). Despite the length, this distance is still shorter than the one predicted for IF*FR, which also has the P2′ Arg out of the binding pocket (**Fig 9F**), corroborating our previous results (**Fig 7A**). This lower preferred interaction is also observed for the two previous

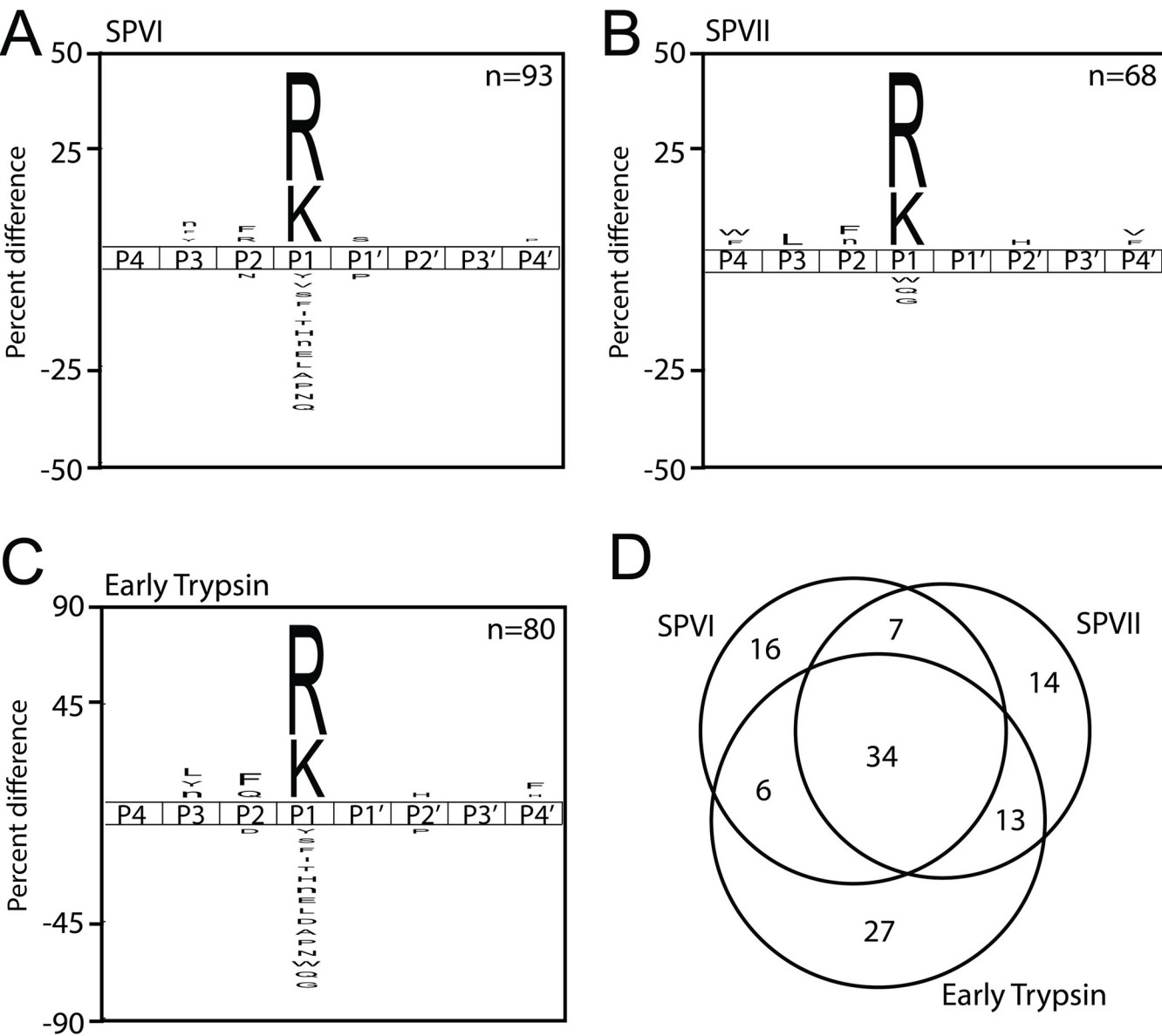

**Fig 6. Substrate specificity profile of trypsin-like serine proteases expressed in the mosquito midgut.** The iceLogo profiles represent the amino acids most frequently found in the P4 to P4′ positions. Cleavage profiles of A) AaSPVI, B) AaSPVII, and C) AaET. Last, D) Venn diagram shows the similarities and differences in substrate specificity among the three enzymes.

substrates QR*VI and YR*MI when predicted to AaLT, which are not within feasible distances for the expected nucleophilic attack (**Fig 9G and 9H**) and corroborate AaLT cleavage profile with no Arg at P1 (**Fig 7A**). Lastly, the Arg at P1′ or P2′ is now predicted to be accommodated on the opposite side of the catalytic serine, differently from the predictions for the Arg at P1 from QR*VI and YR*MI to AaET, AaSPVI, and AaSPVII (**Fig 8**).

## Discussion

The digestion of blood meal proteins by *Ae. aegypti* mosquitoes, which provides essential oligopeptides and amino acid precursors [1,43], is a biphasic process where a network of

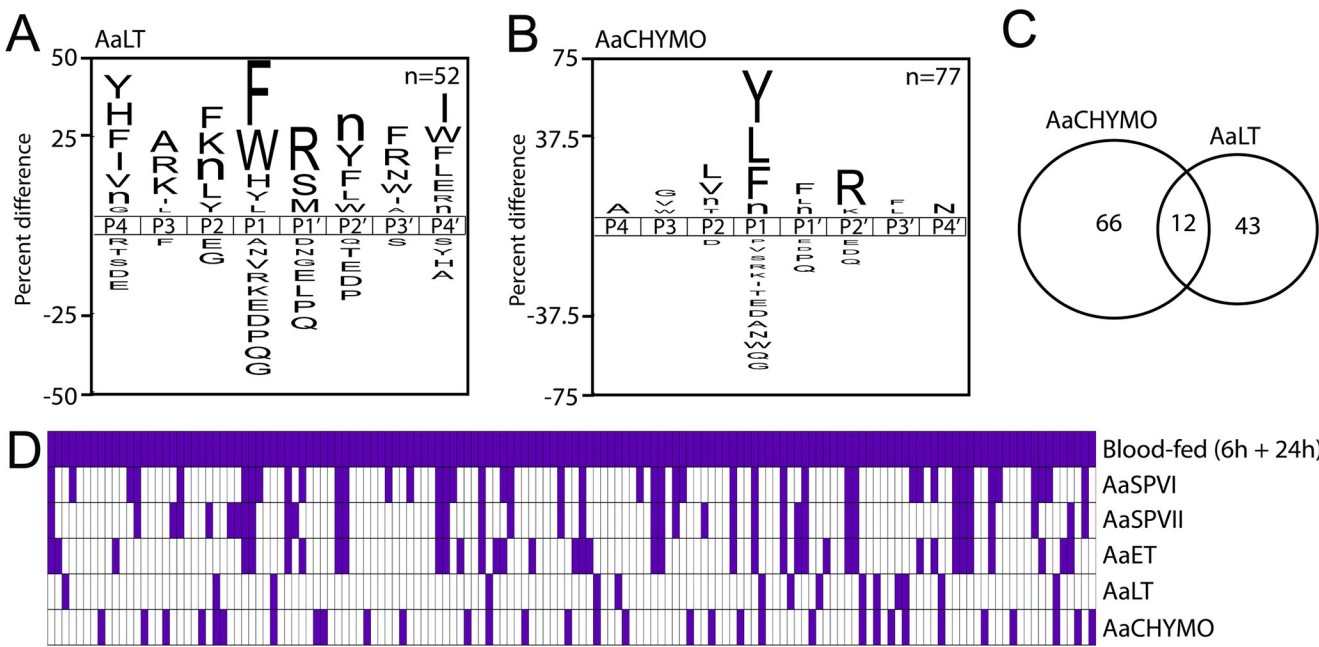

**Fig 7. Substrate specificity profile of AaCHYMO and AaLT serine proteases expressed in the mosquito midgut.** The iceLogo profile represents the amino acids most frequently found at the P4 to P4′ positions. Cleavage profile of A) AaCHYMO and B) AaLT. C) Venn diagram showing the similarities and differences in substrate specificity between AaCHYMO and AaLT. D) Qualitative comparison of cleaved peptides between proteases in blood-fed extracts and the recombinant enzymes. Each column corresponds to a peptide cleavage site. Purple indicates that cleavage was detected, while white indicates not detected.

proteases is released in the midgut: the early phase, leading to an increase in proteolytic activity in the midgut lumen, followed by the late phase, where an even higher amount of proteolytic activity is required to fully digest blood meal proteins. To achieve this, it is proposed that AaET, JHA15, AaCHYMO, and AaSPI are expressed in the early phase (within the first 6 h PBM) to initially digest globular blood meal proteins [1,2,14]. In the late phase of blood meal digestion, AaSPVI, AaSPVII, and AaLT have been shown to be directly involved in blood meal protein digestion, with SPVI and SPVII having trypsin-like activity, while AaLT not having any detectable trypsin-like activity [28]. To further complicate what is known about the biphasic nature of blood meal protein digestion, another set of serine proteases are proposed to be constantly expressed throughout the whole digestion process (termed SPII-SPV) [14]. The mRNA and protein levels of these set of proteases (SPII-SPV) are relatively much lower than AaET, JHA15, AaCHYMO, AaSPI, AaSPVI, AaSPVII, and AaLT [14].

To fully understand this complex blood meal digestion process, we generated a global proteolytic profile of *Ae. aegypti* mosquito midgut extracts before blood feeding (sugar-fed), and 6- and 24-hours post blood feeding (PBF). Sugar-fed midgut extracts were analyzed first and used as a control because these tissues have no detectable amounts of endo-proteolytic activity when using chromogenic reporter substrates, N-benzoyl-arginine-p-nitroanilide (BApNA) or other p-nitroanilide (pNA) substrates [1,12,21,28]. When comparing cleavage preferences at the P4-P4′ amino acid positions between sugar and blood-fed mosquito midgut extracts, a high preference for Arg and Lys at the P1 position was observed for all tissues. This is not surprising given that the major activity in midgut tissue extracts are from trypsin-like proteases, confirming results from previous studies [1,12,28]. However, for sugar-fed extracts, exo-proteolytic activity from mono-aminopeptidases and carboxypeptidases was observed, with a shift from exo-proteolytic activity to endo-proteolytic activity in blood-fed samples. From an

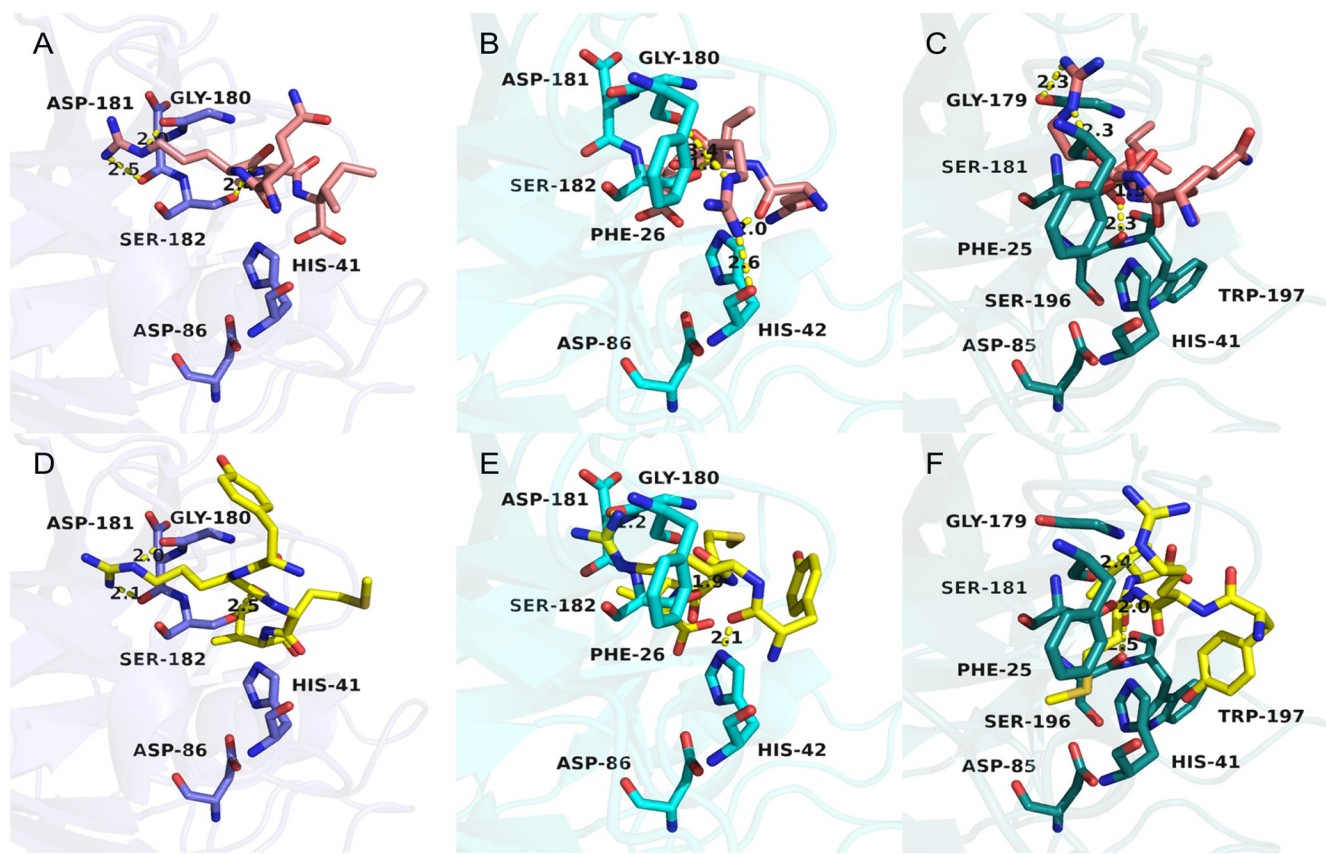

| Enzyme | Substrate (P2-P1*P1'-P2') | Nucleophile distance (Å) | Cleave feasibility (≤ 2.5 Å) | Predicted P1 contacts |
|---|---|---|---|---|
| AaET | QR*VI | 2.4 | Yes | $Gly_{180}$ and $Asp_{181}$ |
| AaSPVI | QR*VI | 1.8 | Yes | $Phe_{26}$ and $His_{42}$ |
| AaSPVII | QR*VI | 1.5 | Yes | $Phe_{25}$ and $Gly_{179}$ |
| AaET | YR*MI | 2.5 | Yes | $Gly_{180}$ and $Asp_{181}$ |
| AaSPVI | YR*MI | 1.9 | Yes | $Gly_{180}$ |
| AaSPVII | YR*MI | 2.0 | Yes | $Phe_{25}$ |

**Fig 8. Substrates having a P1 arginine are predicted to interact with AaET, AaSPVI, and AaSPVII.** Predicted docking poses of the substrate QR*VI (pink) to A) AaET (slate blue), B) AaSPVI (cyan), and C) AaSPVII (teal) predict a similar preference for the P1 arginine close to the catalytic serine, regardless of the P2 position. Similar poses are predicted for YR*MI (yellow) to D) AaET (slate blue), E) AaSPVI (cyan), and F) AaSPVII (teal). P1 is less likely to interact with aspartate or glycine residues in AaSPVII, rather having potential contacts with $Phe_{25}$ and an additional P2' contact with $Trp_{197}$. The expected scissile bond (carbonyl) is predicted to be prone to nucleophile attack, within feasible average distance (≤ 2.5 Å). Important residues are displayed as sticks and labeled (black). Hydrogens were hidden for clarity. Predicted interactions or measured distances (Å) are shown as yellow dashed lines. The table summarizes potential P1 contacts, distances, and cleavage feasibility. All images were generated with PyMOL software (v2.5.7).

evolutionary perspective, this is consistent given that the amino acid content of the three major blood proteins (serum albumin, hemoglobin, and immunoglobulin) contain several Arg and Lys residues. For instance, in human serum albumin, the mature form of albumin, which is responsible for transporting fatty acids in blood, contains 24 Arg residues and 59 Lys residues [44–46]. In addition, the alpha and beta subunit of human hemoglobin each contain three Arg and 11 Lys residues, respectively, for a total of six Arg and 22 Lys residues, given that

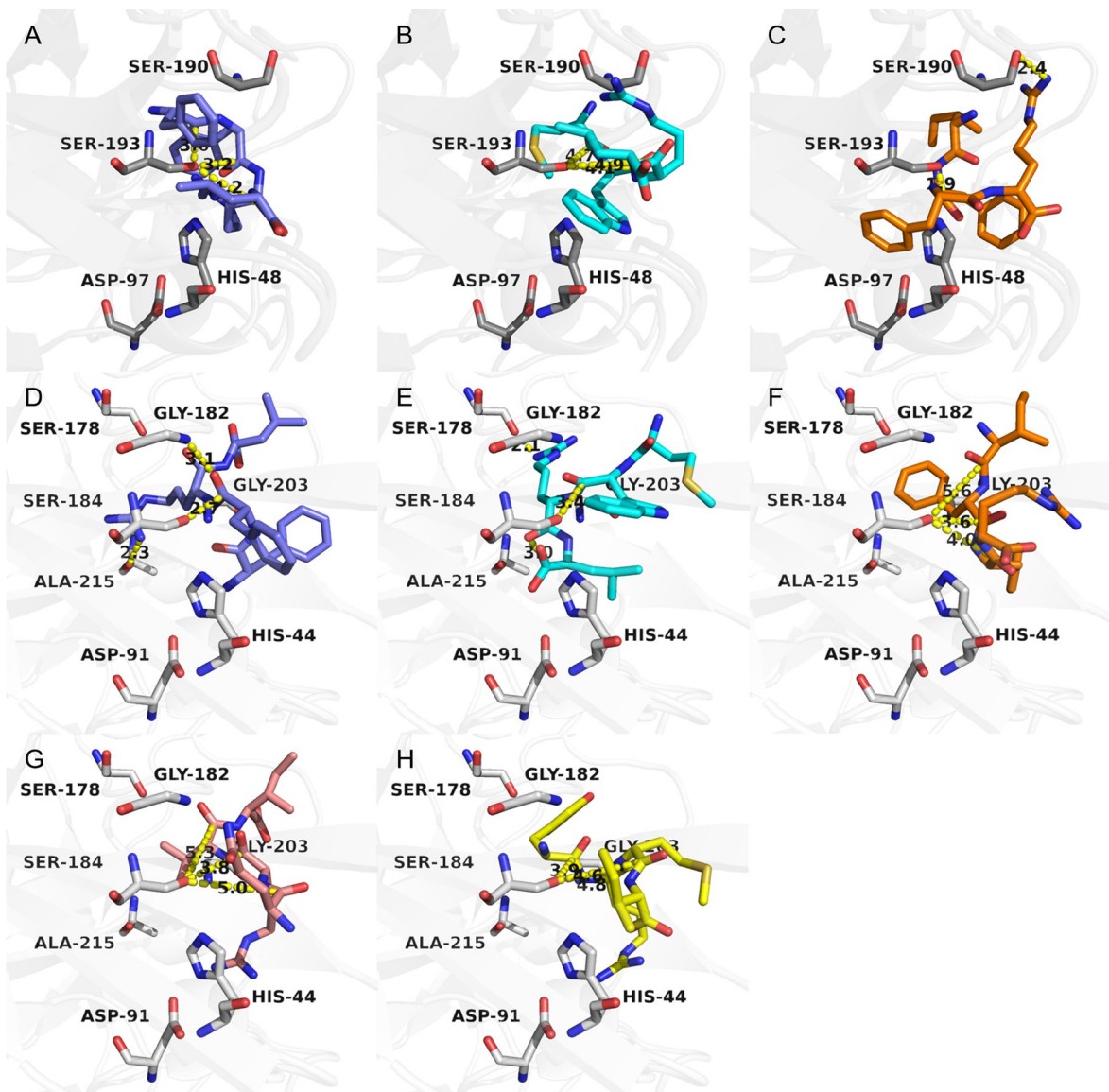

| Enzyme | Substrate (P2-P1*P1′-P2′) | Nucleophile distance (Å) | Cleave feasibility (≤ 2.5 Å) | Predicted P1 or P1′ contacts |
|---|---|---|---|---|
| AaCHYMO | FF*RL | 3.2 | No | - |
| AaCHYMO | MW*RL | 4.1 | No | - |
| AaCHYMO | IF*FR | 1.9 | Yes | $His_{48}$ (P1) |
| AaLT | FF*RL | 2.5 | Yes | $Gly_{182}$ (P1′) |
| AaLT | MW*RL | 3.4 | No | $Ser_{178}$ (P1′) |
| AaLT | IF*FR | 3.6 | No | - |
| AaLT | QR*VI | 3.8 | No | - |
| AaLT | YR*MI | 4.8 | No | - |

**Fig 9. Substrates having Phe at P1′ are predicted to be preferred by AaCHYMO, while AaLT prefers Arg at P1′.** Predicted poses of the substrates A) FF*RL (slate blue) and B) MW*RL (cyan) to AaCHYMO (gray) are not within feasible distances to nucleophilic attack. On the other hand, C) IF*FR (orange) is predicted to have the arginine at P2′ interacting with $Ser_{190}$. Conversely, D) FF*RL (slate blue) and E) MW*RL (cyan) are predicted to interact similarly in AaLT (white), despite the latter being less feasible to nucleophilic attack (~3.4 Å) and having the arginine at P1′ interacting with $Ser_{178}$. Despite the length, this distance is still shorter than the one predicted for IF*FR (orange) in

AaLT. The expected scissile bonds (carbonyl), except for MWRL, are predicted to be prone to nucleophilic attack, within feasible average distance ($\leq 2.5$ Å). This is not predicted to the substrates G) QR*VI (pink) and H) YR*MI (yellow) to AaLT ($\geq 3.9$ Å). Important residues are displayed as sticks and labeled (black). Hydrogens were hidden for clarity. Predicted interactions or measured distances (Å) are shown as yellow dashed lines. The table summarizes potential P1 or P1′ contacts, distances, and cleavage feasibility. All images were generated with PyMOL software (v2.5.7).

hemoglobin is composed of two alpha and two beta subunits [47,48]. As for immunoglobulins, the most abundant form found in human serum is IgG, which is composed of two heavy chains with 13 Arg and 34 Lys in each chain, and two light chains with 14 Arg and 11 Lys amino acid residues in each chain, totaling 27 Arg and 45 Lys residues in the whole protein complex [49–51]. When looking at the crystal structures of each of the blood meal proteins, serum albumin (PDB: 1E7H) [52], hemoglobin (PDB: 1GZX) [47], and IgG (PDB: 1HZH) [49], many Arg and Lys residues are found either in loops or in exposed regions that are accessible to mosquito midgut proteases, which can in turn initiate the proteolytic degradation of these proteins. A previous study has shown degradation activity of serum albumin and hemoglobin globular proteins by AaET, AaSPVI, AaSPVII, and AaLT [28]. This is beneficial because the mosquito must degrade the globular proteins into smaller polypeptides and oligopeptides that can be further processed by other midgut endo-proteases, and eventually amino- and carboxypeptidases to release individual amino acids that can be used for egg production and egg lipid synthesis [1,14,43].

In sugar-fed extracts, most of the detectable protease activity comes from exo-proteolytic enzymes. These results are similar to a previous study, where researchers were able to detect aminopeptidase activity in unfed mosquito midgut extracts, but in this case, a drop in aminopeptidase activity was observed in the 20–24 h PBF samples [53]. In our study, a reduction in aminopeptidase activity at the 24 h timepoint was observed, but with a switch to an increase in carboxypeptidase activity at the same timepoint. This followed with a shift in overall endopeptidase activity in blood fed midgut extracts. These results are expected given that the mosquito must digest blood meal proteins within 40 to 48 h to acquire the necessary amino acid nutrients to fuel the egg laying process. Intact globular blood proteins such as serum albumin are still present at 24 and 30 h PBF [1], which require the activity of endo-proteases to degrade the globular proteins into smaller polypeptides and oligopeptides that can then be processed by amino- and carboxypeptidases. Without exopeptidase activity, single amino acid residues would not be released to be used by the mosquito. Furthermore, the difference in aminopeptidase activity between the two studies could be due to the limitations in proteolytic activity using chromogenic substrates versus the more sensitive fluorescent substrates and mass spectrometry approach used in this study [26,54,55].

The activity of midgut extracts was assayed with the Bz-FVR-AMC substrate in the absence and presence of AEBSF (an irreversible serine protease inhibitor). In the absence of the inhibitor, trypsin-like activity was observed from the 6 h and 24 h PBM midgut extracts, but no activity observed in sugar-fed extracts. This is in line with previous studies that have shown that blood meal digestion in the mosquito midgut is a biphasic process with an initial increase in trypsin-like activity in the first 15 h PBM, followed by a larger increase in trypsin-like activity in the late phase [1]. More importantly, no trypsin-like activity was detected in human blood only samples, which further indicates that the activity is coming specifically from mosquito midgut proteases. Additionally, because no detectable trypsin-like activity was detected from the blood only samples, there was no need to conduct MSP-MS analysis. In the presence of AEBSF, proteolytic activity was eliminated, given the fact that all known midgut proteases belong to the serine protease family. This is important because when comparing trypsin-like activity from wild-type and dsRNA knockdowns to the FLUC control, the knockdown of

AaET, AaLT, AaSPVI, and AaSPVII (the most abundant midgut proteases (based on mRNA and protein expression profiles)), only AaSPVI had the largest effect on trypsin-like activity, with residual trypsin activity still being detected. These results correlate with the 2009 knock-down studies of AaSPVI, AaSPVII, and AaLT, where only the knockdown of AaSPVI had a significant effect on proteolytic activity in the late phase [1]. RNAi knockdown alone did not have a huge effect on trypsin-like activity in the 2009 study, and knockdown effects on fecundity were not completely eliminated even when all three major late phase proteases (AaSPVI, AaSPVII, and AaLT) were knocked down simultaneously [1]. Targeting one or a few midgut proteases may not be enough to have an overall effect on fecundity, and thus a broader, yet mosquito-specific serine protease inhibitor that targets all major midgut proteases may be of better use.

MSP-MS was utilized to determine the cleavage and specificity preferences of purified recombinant mosquito midgut proteases (AaET, AaSPVI, AaSPVII, AaLT, and AaCHYMO). Taking this approach, we can begin to delineate the global proteolytic activity of mosquito midgut extracts and begin to match the overall cleavage specificities of the blood-fed midgut extracts observed in **Fig 3**. We confirmed that AaET, AaSPVI, and AaSPVII are true trypsin-like proteases with preference for Arg and Lys at the P1 position. Based on the amino acid content of the blood meal proteins (many Arg and Lys residues), these proteases would readily initiate the processing of globular proteins in their native structures in the midgut lumen, opening them up to their tertiary, secondary, and primary structures for further processing into polypeptides and oligopeptides [28]. AaET would be the first protease to initiate digestion (found in the early phase), followed by AaSPVI and AaSPVII in the late phase. It is important to note that even after 24 h, intact globular blood meal proteins are still present [1], so the need for other trypsin-like proteases in the late phase are needed to fully degrade blood meal proteins.

Recombinant AaLT was previously expressed and purified but its mode of activation was unknown. Thus, the natural propeptide sequence was replaced with a heterologous enterokinase cleavage site to facilitate the activation of the zymogen form to the active mature form [28]. Although the protease was activated and purified, no trypsin-like activity was detected. To confirm that the recombinantly expressed protease was active, digestion assays with bovine serum albumin (BSA) and hemoglobin (Hb) were conducted in the presence of active AaLT [28]. AaLT cleaved BSA and Hb, but not the trypsin substrate, BApNA. Furthermore, amino acid sequence analysis with known proteases from other organisms revealed that AaLT may likely be a serine collagenase. The best homologous hit found at the time was a serine collagenase from the fiddler crab [28]. The fiddler crab protease is a true serine collagenase being able to cleave collagen and was also shown to have multiple protease specificities from trypsin to elastase [56–58]. With the improvement of producing soluble recombinant AaLT [12], we were able to activate the natural zymogen form with AaSPVI. After purifying and isolating active AaLT, cleavage specificity of AaLT was determined using MSP-MS confirming that this mosquito protease is not a typical trypsin-like enzyme. When looking at the iceLogo analysis, Arg and Lys are two residues that are not preferred at the P1 position. In fact, the preference for this protease is Phe and Trp, with His, Tyr, and Leu also being other amino acids preferred at the P1 site. In addition, of all the proteases analyzed with MSP-MS, AaLT is the only protease that can accommodate a wide range of different amino acids in each of the P4-P4′ positions, indicating the possibility of having promiscuous activity towards several different protein substrates. As indicated, female mosquitoes must digest the blood meal within 40 to 48 h, and therefore it would be advantageous to have diverged a unique protease that could accommodate different amino acids to fully digest and process the blood meal proteins along with other proteases that have high specificity. AaLT is expressed in the late phase, with

protease expression observed starting at 18 h PBM all the way to 36 h PBM [1]. Given that Phe and Trp are the preferred amino acids in AaLT, this preference is similar to chymotrypsin-like enzymes that can accommodate Phe, Tyr, Trp, and to a certain extent Leu [41,42]. In fact, in this study, we confirmed AaCHYMO to be a chymotrypsin-like protease, with preference for Tyr, Leu, Phe, and to a certain extent Met. However, to validate these results, both AaLT and AaCHYMO must be further characterized to confirm the cleavage specificities described here, especially to determine if AaLT can cleave collagen, as predicted by amino acid sequence alignment [28].

Lastly, we aimed to understand how each midgut protease would interact with the shortened peptide sequences by predicting their potential interactions using molecular docking simulations. Our simulations supported the enzyme cleavage preferences observed from our MSP-MS results, such as the predicted P1 Arg contacts in AaET, AaSPVI, and AaSPVII, as shown in **Fig 8**. These P1 interactions mainly happened with aspartate or glycine residues in all three enzymes, favoring the P1*P1′ cleavage site position accommodating close to the catalytic serine for nucleophilic reaction. Such P1 Arg contacts may be favored due to the presence of its five hydrogen bond donors when compared to other residues (*e.g.*, lysine) [59], and arguably benefit from the flexibility of the glycine [60] near the catalytic serine. Conversely, when analyzing AaLT and AaCHYMO in our simulations, the results supported AaCHYMO not tolerating Arg residues at P1′, while preferring Phe instead, as was demonstrated by our biochemical data. This is consistent with previous findings of chymotrypsin-like enzymes, which can accommodate Phe residues [41,42,60]. On the other hand, P1′ Arg was predicted to be preferred by AaLT then P1′ Phe, P1 Arg and P1′ Val, and P1 Arg and P1′ Met, also corroborating our MSP-MS results. One could also argue that these predicted variations may be consistent with the differences between these two enzymes, one being a chymotrypsin-like [41,42] and the other with suggested serine collagenase-like activity [28].

## Conclusions and future directions

Taken together, these results show that MSP-MS is a valuable tool to help elucidate the complexity of the blood meal digestion process in the *Ae. aegypti* mosquito. Global proteolytic profiles of sugar- and blood-fed extracts reveal important specificities and activities that can be mapped by determining the cleavage and specificity of recombinant midgut proteases, which were also supported by our docking simulation studies. By understanding the individual roles of each midgut protease, a better and more specific inhibitor can be developed to achieve the goal of affecting the rate of midgut digestion and vitellogenin synthesis in the fat body, and thus, overall fecundity. Known roles of AaET, AaSPVI, AaSPVII were confirmed using MSP-MS in this study, and more importantly, helped reveal the cleavage specificity of AaLT. This protease, as well as AaCHYMO and the other identified midgut proteases (JHA15, AaSPI, AaSPII-AaSPV) in *Ae. aegypti* are currently being studied to determine their specific roles and activity in the blood meal digestion process. This is the first step in better understanding *Ae. aegypti* blood meal digestion and validating whether this process might be an excellent target for protease inhibitor design.

The overarching goal is the development a new vector control strategy that specifically focuses on *Ae. aegypti* mosquito midgut proteases. In elucidating the activity, and direct role of all mosquito proteolytic enzymes in fecundity, midgut proteases may be a viable target. With the MSP-MS approach and molecular docking studies, along with structural determination, we can design very specific mosquito midgut protease inhibitors with preferred amino acids at the P1-P4 and P1'-P4' sites. This will eliminate non-specific binding to other insect or even mammalian proteases. If successful inhibitors are synthesized, to help deliver these inhibitors

to the mosquito midgut, the attractive toxic sugar bait (ATSB) approach can be taken [61]. ATSB exploits the need of a sugar meal by both male and female mosquitoes. Adult mosquitoes after eclosion have depleted energy stores and must replenish them either through sugar or blood feeding [61–63]. By adding a toxin to the sugar bait (in this case the protease inhibitor) when the mosquitoes feed the inhibitor will end up in the midgut. Additionally, with advancements in nanoparticle engineering for drug delivery [64], protease inhibitors could be encapsulated in a nanoparticle and released when the mosquito imbibes a bloodmeal. A drastic change in pH and midgut contents can help facilitate the release of the inhibitors. Alternatively, if the inhibitor approach does not work, delivery of dsRNA through ATSB is another possibility, which has shown to be successful [61,62].

## Supporting information

**S1 File. MSP-MS data of cleaved products by mosquito midgut extracts and recombinant proteases.**
(XLSX)

**S2 File. Raw data used to obtain Figs 2–4 and 6–7.**
(XLSX)

## Acknowledgments

AAR would like to thank the Department of Chemistry at San Jose State University for allowing him to start his independent career in mosquito protease work. In addition, the Rascón lab would like to thank all his former SJSU research students and the first Chem 131B (Biochemistry Capstone Lab Course) Fall 2013 students for helping with the initial design and cloning of the mosquito midgut protease genes.

## Author Contributions

**Conceptualization:** Anthony J. O'Donoghue, Jun Isoe, Alberto A. Rascón, Jr.

**Data curation:** Anthony J. O'Donoghue, Alberto A. Rascón, Jr.

**Formal analysis:** Anthony J. O'Donoghue, Chenxi Liu, Mateus Sá M. Serafim, Zhenze Jiang, Jun Isoe, Alberto A. Rascón, Jr.

**Funding acquisition:** Alberto A. Rascón, Jr.

**Investigation:** Anthony J. O'Donoghue, Chenxi Liu, Carter J. Simington, Saira Montermoso, Elizabeth Moreno-Galvez, Mateus Sá M. Serafim, Olive E. Burata, Rachael M. Lucero, James T. Nguyen, Daniel Fong, Khanh Tran, Neomi Millan, Jamie M. Gallimore, Kamille Parungao, Jonathan Fong, Brian M. Suzuki, Zhenze Jiang, Jun Isoe, Alberto A. Rascón, Jr.

**Methodology:** Anthony J. O'Donoghue, Mateus Sá M. Serafim, Jun Isoe, Alberto A. Rascón, Jr.

**Project administration:** Anthony J. O'Donoghue, Jun Isoe, Alberto A. Rascón, Jr.

**Resources:** Anthony J. O'Donoghue, Jun Isoe, Alberto A. Rascón, Jr.

**Supervision:** Anthony J. O'Donoghue, Alberto A. Rascón, Jr.

**Validation:** Anthony J. O'Donoghue, Mateus Sá M. Serafim, Jun Isoe, Alberto A. Rascón, Jr.

**Visualization:** Anthony J. O'Donoghue, Mateus Sá M. Serafim, Jun Isoe, Alberto A. Rascón, Jr.

**Writing – original draft:** Alberto A. Rascón, Jr.

**Writing – review & editing:** Anthony J. O'Donoghue, Mateus Sá M. Serafim, Jun Isoe, Alberto A. Rascón, Jr.

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
