## [Decision Letter · Decision Letter 0]

27 Nov 2024

PNTD-D-24-01354Comprehensive proteolytic profiling of Aedes aegypti mosquito midgut extracts: Unraveling the blood meal protein digestion systemPLOS Neglected Tropical Diseases

Dear Dr. Rascón, Jr.,

Thank you for submitting your manuscript to PLOS Neglected Tropical Diseases. After careful consideration, we feel that it has merit but does not fully meet PLOS Neglected Tropical Diseases's publication criteria as it currently stands. Therefore, we invite you to submit a revised version of the manuscript that addresses the points raised during the review process.

Please submit your revised manuscript within 60 days Jan 26 2025 11:59PM. If you will need more time than this to complete your revisions, please reply to this message or contact the journal office at plosntds@plos.org. Please include the following items when submitting your revised manuscript:

*
A rebuttal letter that responds to each point raised by the editor and reviewer(s). You should upload this letter as a separate file labeled 'Response to Reviewers'. This file does not need to include responses to any formatting updates and technical items listed in the 'Journal Requirements' section below. *
A marked-up copy of your manuscript that highlights changes made to the original version. You should upload this as a separate file labeled 'Revised Manuscript with Track Changes'. *
An unmarked version of your revised paper without tracked changes. You should upload this as a separate file labeled 'Manuscript'. If you would like to make changes to your financial disclosure, competing interests statement, or data availability statement, please make these updates within the submission form at the time of resubmission. Guidelines for resubmitting your figure files are available below the reviewer comments at the end of this letter. We look forward to receiving your revised manuscript. Kind regards, R. Manjunatha KiniAcademic EditorPLOS Neglected Tropical Diseases

Nigel BeebeSection EditorPLOS Neglected Tropical Diseases 

Shaden Kamhawi

co-Editor-in-Chief

Paul Brindley

co-Editor-in-Chief

**Additional Editor Comments:**

The manuscript describes the proteolytic site specificity of mosquito midgut proteases. The authors have provided some details of their strategies, experimental design and results to support their conclusions. Both reviewers have carefully evaluated the manuscript and requested revisions to the manuscript.

Editor's comments (in addition to the reviewers' comments):

The authors are planning to design highly selective protease inhibitors to inhibit the mosquito midgut proteases to reduce fecundity. It is not clear how such inhibitors be delivered to mosquitoes as the inhibitors have to be in human blood. The authors should add a short paragraph on how such inhibitors be used to reduce fecundity.

By docking studies, the authors should also identify the residues that interact with P1 and P1' residues that define the substrate binding pockets and cleavage site selectivity.  **Journal Requirements:**

At this stage, the following Authors/Authors require contributions: Anthony J. O’Donoghue, Chenxi Lui, Carter J. Simington, Saira Montermoso, Elizabeth Moreno-Galvez, Mateus Sá M Serafim, Olive E. Burata, Rachael M. Lucero, James T. Nguyen, Daniel Fong, Khanh Tran, Neomi Millan, Jamie M. Gallimore, Kamille A. Parungao, Jonathan Fong, Brian M. Suzuki, Zhenze Jiang, Jun Isoe, and Alberto Amado Rascón, Jr.. Please ensure that the full contributions of each author are acknowledged in the "Add/Edit/Remove Authors" section of our submission form.

- ® on pages: 10, 11, and 21.

- TM on pages: 11, 15, and 22..

Potential Copyright Issues:

i) Please confirm (a) that you are the photographer of Figure 1B, and and 1D., or (b) provide written permission from the photographer to publish the photo(s) under our CC BY 4.0 license.

ii) Figure 1. Please confirm whether you drew the images / clip-art within the figure panels by hand. If you did not draw the images, please provide (a) a link to the source of the images or icons and their license / terms of use; or (b) written permission from the copyright holder to publish the images or icons under our CC BY 4.0 license. Alternatively, you may replace the images with open source alternatives. See these open source resources you may use to replace images / clip-art:

6) Please amend your detailed Financial Disclosure statement. This is published with the article. It must therefore be completed in full sentences and contain the exact wording you wish to be published. Please ensure that the funders and grant numbers match between the Financial Disclosure field and the Funding Information tab in your submission form. Note that the funders must be provided in the same order in both places as well.

  **Reviewers' Comments:**

Reviewer's Responses to Questions

**Key Review Criteria Required for Acceptance?**

**Methods**

-Are the objectives of the study clearly articulated with a clear testable hypothesis stated?

-Is the study design appropriate to address the stated objectives?

-Is the population clearly described and appropriate for the hypothesis being tested?

-Is the sample size sufficient to ensure adequate power to address the hypothesis being tested?

-Were correct statistical analysis used to support conclusions?

-Are there concerns about ethical or regulatory requirements being met?

Reviewer #1: The reviewer only considers one major critique about the design of this study. The major concern is that the authors did not employ statistical analyses to support their conclusions? This critique covers parts of figures 2, 3 and 4.

For Fig. 2B&C, the authors need to clearly/ define what the error bars signify (SDev, SEM ...etc)?

For Fig. 3C, are the differences in enzymatic activity between sugar and blood-fed samples significant?

For Figure 3D, the authors did not clearly/ define what the error bars signify (SDev, SEM ...etc)? Why or why not are the differences in Peak Area between 6h and 24h blood-fed significant ?

For Fig. 3G, If the graphs were from 4-techical replicates, why no error boars from the measure of fluorescence signal?

Fig. 4 A&B, No control for AEBSF baseline activity (i.e non-fed conditions). What are the error bars? SDev, SEM ...etc. how many replicates? For which RNAi knock-downs are the differences significant?

Reviewer #2: The study by O’Donoghue et al. profiled the midgut proteolytic activity of Aedes aegypti using Multiplex Substrate Profiling by Mass Spectrometry (MSP-MS), recombinant expression and characterization of recombinant midgut proteases, and molecular docking simulations. The objectives of the study were clear (characterizing substrate specificity of Ae. aegypti midgut proteases), the methods were appropriate, and the objective impressively addressed through multiple approaches.

However, there is detail missing in the described methodology that will need to be provided, in particular there is not much information given regarding midgut sample numbers and additional details regarding the library of designed peptides is needed. It would have good to have seen a blood only control (no mosquito midguts) with the MSP-MS.

**Results**

-Does the analysis presented match the analysis plan?

-Are the results clearly and completely presented?

-Are the figures (Tables, Images) of sufficient quality for clarity?

Reviewer #1: The reviewer agrees that the figures are of sufficient quality, with the suggestion that the authors used a larger font size in the figure graphs. The reviewer suggest that the authors add the legends to Figure 3C like the ones on Figure 3D and 3G for to be consistent.

Reviewer #2: Results were clearly presented and figures of sufficient quality for publication. My only comment would be that for the molecular docking studies, it would be good to have a table that displays the distances, highlighting which distances were considered possible for nucleophile attack. The molecular docking simulation results were a bit vague.

**Conclusions**

-Are the conclusions supported by the data presented?

-Are the limitations of analysis clearly described?

-Do the authors discuss how these data can be helpful to advance our understanding of the topic under study?

-Is public health relevance addressed?

Reviewer #1: The authors addressed the public health relevance.

This study is a technique-driven/descriptive study of Ae. aeqypti gut serine proteases involved in the midgut. The authors data (with the inclusion of statistics) support the conclusions of their study that trypsin-like proteases are the main activity in the mosquito midgut.

The authors discuss how the data and methodology opens doors to novel strategies for controlling Ae. aegypti.

Reviewer #2: Conclusions are well support, and the research question has been addressed with multiple approaches (MSP-MS, recombinant expression of proteases, and molecular docking simulations). It would be good to add a bit more to the ‘Conclusions and Future Directions’ that more specifically describes the next steps.

**Editorial and Data Presentation Modifications?**

Reviewer #1: The reviewer has one minor critique about the presentation of the Results and Discussion sections. The Results and Discussion sections were stylistically inefficient because the authors began most paragraphs by rewriting or re-integrating a method into it. Furthermore, the Discussion section re-chronicled the Results figures. The Discussion section needs to be shorter and to the point.

Reviewer #2: Minor revision.The manuscript is well written with high quality figures. There are some additional details missing from the methods that will need to be added before publication.

**Summary and General Comments**

Reviewer #1: The reviewer has one minor critique about the presentation of the Results and Discussion sections. The Results and Discussion sections were stylistically inefficient because the authors began most paragraphs by rewriting or re-integrating a method into it. Furthermore, the Discussion section re-chronicled the Results figures. The Discussion section needs to be shorter and to the point.

Reviewer #2: O’Donoghue et al. details an impressive study profiling the substrate specificity of Ae. aegytpi midgut proteases by mass spectrometry, functional characterization of recombinant proteases, and molecular docking simulations. There is detail missing in the methods and support/justifications required for some results, please find comments below:

Methods:

Line 174: “maintained on 10% sucrose until needed for experiments”. How long of a period was this? Was successful knockdown confirmed with qPCR? What percentage of knockdown was achieved? Was a control used? How many mosquitoes were used?

Line 179: “the tetradecapeptide library consists of 228 rationally designed peptides” Could more information be provided here? Did the authors design these peptides or was this a commercially available synthetic library?

Line 185: How many midgut samples? Was the whole synthetic library incubated with human blood (no midgut extracts) as a control?

I do not see where the mass spectrometry data has been made publicly available (MassIVE, Figshare, etc.). This is a requirement of PLOS Data policy.

Results:

I feel it is a bit unclear on the reasoning behind why these proteases were picked for recombinant expression. More detail should be provided regarding the abundance of these proteases in the midgut (I assume they were picked because they are highly abundant in transcriptomics or proteomics data).

What distance is considered possible for nucleophile attack? More details needed for the criteria used. It would be nice to see a table with all these distances.

Discussion:

Line 599: What do we know about expression and abundance of these proteases during blood feeding? That should be incorporated into the discussion.

PLOS authors have the option to publish the peer review history of their article (what does this mean?). If published, this will include your full peer review and any attached files.

Reviewer #1: No

Reviewer #2: No

---

## [Decision Letter · Decision Letter 1]

28 Jan 2025

Dear Dr. Rascón, Jr.,

We are pleased to inform you that your manuscript 'Comprehensive proteolytic profiling of Aedes aegypti mosquito midgut extracts: Unraveling the blood meal protein digestion system' has been provisionally accepted for publication in PLOS Neglected Tropical Diseases.

Best regards,

R. Manjunatha Kini

Academic Editor

Nigel Beebe

Section Editor

Shaden Kamhawi

co-Editor-in-Chief

Paul Brindley

co-Editor-in-Chief

The authors have satisfactorily addressed the questions /queries raised by the reviewers and the editor.

Reviewer's Responses to Questions

**Key Review Criteria Required for Acceptance?**

**Methods**

-Are the objectives of the study clearly articulated with a clear testable hypothesis stated?

-Is the study design appropriate to address the stated objectives?

-Is the population clearly described and appropriate for the hypothesis being tested?

-Is the sample size sufficient to ensure adequate power to address the hypothesis being tested?

-Were correct statistical analysis used to support conclusions?

-Are there concerns about ethical or regulatory requirements being met?

Reviewer #1: The authors addressed the major critique from this review by employing basic statistical analyses to support their data.

Reviewer #2: I am satisfied with the manuscript revision and point-by-point responses by the authors. The previously missing detail in the methods has now been added.

**Results**

-Does the analysis presented match the analysis plan?

-Are the results clearly and completely presented?

-Are the figures (Tables, Images) of sufficient quality for clarity?

Reviewer #1: The authors have added legends to all figures as recommended by this reviewer.

Reviewer #2: All of my previous comments on the results have been addressed.

**Conclusions**

-Are the conclusions supported by the data presented?

-Are the limitations of analysis clearly described?

-Do the authors discuss how these data can be helpful to advance our understanding of the topic under study?

-Is public health relevance addressed?

Reviewer #1: The authors data better supports the conclusions of their study that trypsin-like proteases are the main activity in the mosquito midgut.

Reviewer #2: I am satisfied with the manuscript conclusions.

**Editorial and Data Presentation Modifications?**

Reviewer #1: No further data modifications are needed.

Reviewer #2: Before publication, I would make sure to check the resolution of all figures. It is a bit difficult to read the amino acid residue labels in the structural figures (Fig 8 and 9), but I am sure this will be adjusted to make sure all figures are publication quality.

**Summary and General Comments**

Reviewer #1: Minor critiques:

Line 699 lists AaSPVI twice (should one be AaSPVII?). The authors conclude that more specific serine protease inhibitors may be of better use. However, the data suggests that global/non-specific serine protease inhibitors like AEBSF would be the better tools. Do the authors hypothesize that inhibiting all midgut serine protease activity to abrogate fecundity in mosquitoes is necessary because of their built-in redundancy? The reader would be interested to know about the hypothetical effects that AEBSF would have on protein digestion or fecundity if ingested by mosquitoes or injected into midgut. Are there strengths or weaknesses in combining MSP-MS with AEBSF or other global inhibitors to discover midgut proteases that are key to eliminating mosquito fecundity?

Reviewer #2: This manuscript covers an impressive amount of experimental work and all my comments have been addressed.

PLOS authors have the option to publish the peer review history of their article (what does this mean?). If published, this will include your full peer review and any attached files.

Reviewer #1: No

Reviewer #2: No

---

## [Editor Report · Acceptance letter]

2 Feb 2025

Dear Dr. Rascón, Jr.,

We are delighted to inform you that your manuscript, "Comprehensive proteolytic profiling of Aedes aegypti mosquito midgut extracts: Unraveling the blood meal protein digestion system," has been formally accepted for publication in PLOS Neglected Tropical Diseases.

Best regards,

Shaden Kamhawi

co-Editor-in-Chief

Paul Brindley

co-Editor-in-Chief
